
# Aerosol and VOC emission factor measurements for African anthropogenic sources

Sekou Keita[1], Cathy Liousse[2], Véronique Yoboué[1], Pamela Dominutti[3], Benjamin Guinot[2], E-Michel Assamoi[1], Agnes Borbon[3], Sophie L. Haslett[4], Laetitia Bouvier[3], Aurélie Colomb[3], Hugh Coe[4], Aristide Akpo[5], Jacques Adon[2], Julien Bahino[1], Madina Doumbia[1], Julien Djossou[5], Corinne Galy-Lacaux[2], Eric Gardrat[2], Sylvain Gnamien[1], Jean F. Léon[2], Money Ossohou[1], E. Touré N'Datchoh[1], Laurent Roblou[2]

[1]Laboratoire de Physique de l'Atmosphère, Université Félix Houphouët-Boigny, Abidjan, BPV 34, Côte d'Ivoire
[2]Laboratoire d'Aérologie, Université de Toulouse, CNRS, UPS, France
[3]Université Clermont Auvergne, CNRS, LaMP, F-63000 Clermont-Ferrand, France
[4]Centre for Atmospheric Science, University of Manchester, Manchester, M13 9PL, United Kingdom
[5]Laboratoire de Physique du Rayonnement, Université d'Abomey Calavi, Benin

*Correspondence to*: Sekou Keita (sekkeith@yahoo.fr)

**Abstract.**

A number of campaigns have been carried out to establish the emission factors of pollutants from fuel combustion in West Africa, as part of work package 2 ('Air Pollution and Health') of the DACCIWA (Dynamics-Aerosol-Chemistry-Cloud Interactions in West Africa) FP7 program. Emission sources considered here include wood and charcoal burning, charcoal making, open waste burning, and vehicles including trucks, cars, buses and two-wheeled vehicles. Emission factors of total particulate matter, black carbon, primary organic carbon and non-methane volatile organic compounds (NMVOC) have been established. In addition, emission factor measurements were performed in combustion chambers in order to reproduce field burning conditions for tropical hardwood, and obtain particulate emission factors by size (PM0.25, PM1, PM2.5 and PM10). Aerosol samples were collected on quartz filters and analysed using gravimetric and thermal methods. The emission factors of 50 NMVOC species were determined using systematic off-line sampling. Emission factors from wood burning for black carbon, organic carbon and total particulate matter were 0.8 ± 0.4 g/kg of dry matter (dm), 9.29 ± 3.82 g/kg dm and 34.54 ± 20.6 g/kg dm, respectively. From traffic sources, the highest emission factors for all particulate species were emitted from two wheeled vehicles with two-stroke engines (2.74 g/kg fuel for black carbon, 65.11 g/kg fuel for organic carbon and 496 g/kg fuel for total particulate matter). The emissions of NMVOCs were lower than those of particles for all sources aside from traffic. The largest NMVOC emissions were observed for two-stroke two-wheeled vehicles, which were up to three times higher than emissions from light-duty and heavy-duty vehicles. Isoprene and monoterpenes, which are usually associated with biogenic emissions, were present in almost all anthropogenic source categories and could be as significant as aromatic emissions in wood burning (1 g/kg dm). Black carbon was primarily emitted in the ultrafine fraction, with 77% of the total mass being emitted as particles smaller than 0.25 µm. This study observed higher particle and NMVOC emission factors than





those in the current literature. This study underlines the important role of in-situ measurements in deriving realistic and representative emission factors.

## 1 Introduction

Air pollution and its consequences on air quality, human health and climate are particularly worrying in Africa. First, there is a rich mixture of sources of pollutants: natural sources with Sahelian and Saharan dust emissions combine with anthropogenic sources including biomass burning, traffic, industry, residential cooking, power plant emissions and others. Up to now, dust and biomass burning were considered to be predominant and many studies have been conducted on these sources. However, due to urbanization and population density increases, it has been seen that anthropogenic emissions linked to urban activities

could be as important as the "well-known" sources (Marticorena et al., 2010). Second, up to now, this expected emission increase has not been accompanied by any regulations. If nothing is done sooner, the climate and health impacts could be significant by 2030 (Liousse et al., 2014).

In West Africa, domestic fires (for cooking) and traffic have been identified as the main anthropogenic sources of carbonaceous particle emissions (Liousse et al., 2014; Malavelle et al., 2011; Liousse et al., 2010; Marticorena et al., 2010). Andreae and

Merlet, (2001) have already shown that domestic fires used for cooking are an important source of primary carbon worldwide. Globally, more than three billion people use solid fuels such as charcoal, agricultural residues and wood, which are known to be the main source of energy in households (Anenberg and Bayer, 2013), where more than 90% of this consumption occurred in developing countries (Wang et al., 2013). In Africa, traffic emissions are characterized by an aging fleet (more than 80 % are second-hand vehicles) (Kablan, 2010). Most of these vehicles are more than 20 years old (Ministry of Transport, 2012)

and are as such highly polluting due to inefficient combustion (Boughedaoui et al., 2009). Robert et al., (2007) and Peltier et al., (2011) have shown that traffic vehicle emission factors depend on the type of engine, its maintenance, its age and the fuel it uses, as well as environmental conditions. In some countries in the region, it is also important to note the importance of two-wheeled vehicles (two- stroke or four-stroke engines) using a mixture of oil and gasoline derived from smuggling (Assamoi and Liousse, 2010).

In addition, there are sources that emit very high levels of pollutants such as solid waste burning, which have not been well studied. In most African countries, solid waste collection systems are insufficient, leading some people to choose to remove waste using open burning (Wiedinmyer et al., 2014). In many African countries, open burning occurs at public landfills due to the lack of a modern incinerator.

One approach to quantifying air pollution and impacts, and to formulating emission reduction strategies, is to use atmospheric

modelling tools which require emission inventories as source data (Bond et al., 2004; Junker and Liousse, 2008; Granier et al., 2011; Smith et al., 2011; Klimont et al., 2013). Emission inventories are built based on activity and pollutant emission factor (EF) data. However, none of the above-mentioned sources are well documented.





The existing emission inventories are often global and involve many uncertainties, particularly in Africa (Assamoi, 2010). The emission inventory uncertainties result on one hand, from uncertainties in the activity and emission factor data themselves (Liousse et al., 2014; Bond et al., 2013; Zhang and Tao, 2009; Zhao et al., 2011), and on the other hand from spatial keys used to geographically distribute pollutant emissions. Therefore, the use of local activity data and emission factors derived from

local measurements on Africa-specific sources may help to reduce uncertainties in emission inventories.

In the past, several studies have focused on biomass burning emission factors in Africa. Andreae and Merlet (2001) and Akagi et al. (2011) have conducted studies to compile all available biomass burning EFs for a number of gaseous and particulate species. EF measurements for traffic vehicles in Africa, however, are very rare and sometimes non-existent, which means that even in the existing African Regional Inventory, literature EFs must be used (Liousse et al., 2014). Applying such values to

developing country values is a large source of uncertainties.

Moreover, different methods for measuring EFs may give different results. These differences are related to the sampling method: for example, in the traffic sector, we can cite measurements at the exhaust pipe with portable equipment, measurements using on-board tools in vehicles, measurements in road tunnels, measurements using remote sensing and measurements in the laboratory (e.g combustion chamber, test bench) (Franco et al., 2013). Each of these existing methods of

EF measurement has its strengths and weaknesses. These differences are also due to the different pollutant analysis methods (that is the case for carbonaceous particles). In terms of pollutants, it is necessary to focus on carbonaceous aerosols (organic carbon and black carbon) since carbonaceous aerosols are the main constituents of the particle phase of combustion activity emissions. It is also interesting to study non-methane volatile organic compounds (NMVOC). The emission factors of these components is not well-known despite their expected impact on air quality and climate for Particulate Matter (PM) and on

ozone and secondary organic aerosols (SOA) formation for NMVOC (Matsui et al., 2009; Yokelson et al., 2009; Sharma et al., 2015).

Our study is included in the frame of the DACCIWA (Dynamics-Aerosol-Chemistry-Cloud Interactions in West Africa) programme (Knippertz et al., 2015) within work package 2 (WP2), which focuses on air pollution and health impacts. DACCIWA WP2 has many tasks including one on emission inventories. In this framework, several campaigns to measure EF

were performed.

This work aims to provide a database of EFs for total particulate matter (TPM) mass, black carbon (BC), primary organic carbon (OC) and combustion gases (NMVOC) for pollution sources specific to Africa. The focus was on domestic fires using wood and charcoal mostly for cooking, charcoal making and solid waste disposal by open fires. Emissions related specifically to road traffic include studies on vehicle categories (light duty and heavy duty), energy (gasoline or diesel engine), use (private

and public transport) and age (old and recent).

In section 1, this paper describes the main studied African emissions sources and the methodology used to calculate EFs. Section 2 deals with the results of field EF measurements including a comparison with literature values. In this section, combustion chamber EF results are also added.



## 2 Methodology and materials

Two types of measurements were carried out in the frame of DACCIWA WP2 for emission factor measurement experiments: field measurements for all studied sources and combustion chamber measurements for fuelwood. All EF measurements were carried out in the plume, at 1-1.5 m from the combustion source.

### 5  2.1 Emission factors

Emission factors are defined as the amount of pollutant emitted per kilogram of burned fuel. EFs are determined using the carbon balance method (Ferek et al., 1998, Radke et al., 1988, Ward et al., 1982). The amount of carbon emitted to the atmosphere during combustion and that contained in each fuel allow an estimation of the amount of fuel burnt. The EFs are determined from concentration measurements in the plume and in ambient air outside the emission sources. Previous studies

10  (Hall et al., 2012; Chen et al., 2007; Gupta et al., 2001) have shown that, during fuel combustion, approximately 95% of carbon is emitted into the atmosphere as carbon dioxide ($CO_2$) and carbon monoxide (CO). It is therefore reasonable to estimate the emitted amount of carbon from CO and $CO_2$ concentrations by neglecting hydrocarbons and particles, implying a minor overestimation of EF values (Pant and Harrison, 2013; Hall et al., 2012; Chen et al., 2007; Gupta et al., 2001; Yokelson et al., 2007; Shen et al., 2010; Roden et al., 2006).

$$EF(x) = \frac{\frac{\Delta[X]}{\Delta[CO2]+\Delta[CO]}*MM_x}{12} * f_c * MCE * 10^3 \qquad (Eq1)$$

where EF (x) is the emission factor of the pollutant x in g / kg fuel burnt; $\Delta[x] = [x]_{smoke} - [x]_{background}$ is the mixing ratio of x in fresh smoke plume and background air, respectively (it is noted that ambient concentrations of TPM, BC, OC and NMVOC before combustion are assumed to be zero); MMx is the molar mass of x ($gmol^{-1}$) and 12 is the molar mass of carbon (g.mol$^{-1}$

20  ); MCE is the modified combustion efficiency and $f_c$ is the mass fraction of carbon in the fuel.

The modified combustion efficiency (MCE) is defined as:

$$MCE = \frac{\Delta[CO2]}{\Delta[CO2]+\Delta[CO]} \qquad (Eq2)$$

The MCE depends on the relative importance of the two main phases of combustion: the flaming and smoldering phases. The flaming phase is characterized by very high temperature combustion and oxygenation, and the smoldering phase by low

25  temperature and oxygenation (Ward and Radke, 1993). From laboratory tests on biomass burning, several authors have demonstrated that MCE is around 0.99 for pure flaming (Chen et al., 2007; Yokelson et al., 1996), and varies between 0.65-0.85 for smoldering (Akagi et al., 2011). In this study, an average MCE was determined for each studied source from measurements.





## 2.2 Description of field measurements

Three field campaigns of emission factor measurements were performed as part of DACCIWA WP2. The first in March 2015 in Abidjan, the second in July 2015 in both Abidjan and Cotonou, and the third in July 2016, also in Abidjan.During these campaigns, several sources were studied. (1) Open solid waste burning fires: Eight EF measurements were carried out at
"Akouédo" landfill, the largest (153 ha) and the official landfill site in the east of Abidjan District, in eight measurement points chosen to represent the combustion of waste diversity (dry, wet, old or fresh waste). These measurements were carried out in the ambient atmosphere within a combustion plume located about 1-1.5 m above the source. (2) Charcoal and wood burning fire: Eight samplings were carried out for charcoal burning EFs, including six in Abidjan and two in Cotonou. For wood burning EFs, four measurements were carried out in Abidjan using two different species of tropical African hardwood, Hevea
(hevea brasiliesis) and Iroko (Milicia Excelsa), respectively. These two wood species of tropical hardwood have different characteristics and are mainly used in urban areas for cooking, heating and services (bakeries, power plants, etc.). During these measurements, wood and charcoal were burned in two types of stoves traditionally used in the West African region for cooking, made of metal and of baked earth. These measurements include different phases of combustion (pyrolysis, flaming and smoldering). (3) Charcoal making fire (CHM): Eight tests were carried out on traditional charcoal making furnaces, 3 of these
8 tests were located in the outskirts of Abidjan and 5 tests in a rural area at 2 km far from the Lamto geophysical station (Lamto is 260 km far from Abidjan). The CHM kiln was prepared by charcoal producers who use all types of available dense wood. The kiln was covered with a layer of leaves and a layer of earth of about 10 cm thick. The draught, needed for the propagation of the pyrolysis, comes from an air circulation between the base of the kiln and a row of holes made in a horizontal plane, which are closed when the charcoal producers opens a new row below the previous one. The smoke was sampled at the holes
made in the CHM kiln. (4) Combustion of fossil fuels in the traffic sector: EF measurements were carried out on several vehicles in Abidjan and Cotonou: cars, buses, trucks, mopeds, gasoline and diesel vehicles. Both new (under 10 years) and old vehicles (over 10 years) have been studied. For each type of vehicle, at least two tests were performed, simulating several engine speeds. During these tests, the samples were taken in the plume, 1-1.5 m from the exhaust pipe.

## 2.3 Field measurement and sampling equipment

According to equation (Eq1), it is necessary to quantify the amounts of $CO_2$ and $CO$ emitted to the atmosphere to determine the amount of carbon emitted during the combustion and calculate the MCE. The QTRAK-7575, developed by TSI, was used to measure $CO_2$ and $CO$ gas concentrations. This allows the measurement of real time atmospheric $CO_2$ and $CO$ concentrations. The $CO$ concentration is determined using an electrochemical sensor with a sensitivity of 0 to 500 ppm with $\pm$ 3% accuracy.
The $CO_2$ concentration is measured using a non-dispersive infrared detector with a sensitivity of 0 to 5000 ppm with an accuracy of $\pm$ 3. The difference between $CO_2$ and $CO$ in the fresh smoke plume and the background air allowed the amount of carbon emitted into the atmosphere during each sampling to be obtained. NMVOCs were actively sampled using sorbent tubes



containing multisorbent materials (Perkin-Elmer® and TERA-Environnement), previously conditioned by flowing purified air through them at a rate of 100 mL min⁻¹, for 5 hours at 320 °C using an adsorbent thermal regenerator. Duplicate near-sources measurements were performed for 15 minutes using a manual pump (Accuro 2000, Draeger) with a controlled flow of 100 mL.min⁻¹. The particle collection line consisted of a pump with a flow rate of 9.5 litres per minute (lpm) for sampling all

particle sizes, a volumetric counter for quantifying sampled air volume and a filter holder on which a quartz filter was mounted. Before sampling, the filters were cleaned by heating for 48 hours at 340° C. After sampling they were kept at a temperature of 5° C to avoid any contamination of the samples.

**2.4 Combustion chamber measurements**

Combustion chamber sampling took place in a chamber where plumes from emission sources were conveyed. Therefore,
concentrations were diluted compared to the measurements carried out on ground field at the sources. The tests were conducted in two combustion chambers with different configurations. The combustion chamber of Lannemezan (Guillon et al., 2013) (Laboratoire d'Aérologie, UMR 5560, Toulouse, France) and the combustion chamber at the University of Edinburgh's School of Engineering were used. In Lannemezan, the dark dilution chamber (10m x 4m x 4m) allowed measurement of concentrations at low temperatures, in a homogeneous atmosphere where no photochemistry occurred. The fuels were burned in a stove
connected to the combustion chamber by a chimney 1.5 m high and 15 cm diameter (Figure 1). Two QTRAKs were used to measure $CO_2$ and CO concentrations, temperature and associated relative humidity in the room. The chamber remained closed during all phases of combustion, monitored from an adjacent room to the combustion chamber. After homogenization of the plume within the chamber, five filter sampling lines corresponding respectively to the cut-off heads PM0.25, PM1, PM2.5, PM10 and TPM were used to collected aerosols for 15-25 minutes on average. For each of the five lines, the pumps were
coupled to flow regulators to allow aerosol selection by particle size classes. Between two experiments, the chamber was opened (ventilated) to allow all sensors to return to their background values. Four tests were carried out with Hevea wood and charcoal from Côte d'Ivoire.

Combustion experiments were also conducted using the FM-Global Fire Propagation Apparatus (FPA) at the Edinburgh University School of Engineering facility. The FPA allows the burning of small samples of fuel under controlled conditions
(Brohez et al., 2006). The sample holder was placed on a mass balance that provides sample mass evolution during the experiment. The samples were surrounded by four infrared lamps, irradiating uniformly at 30 kWm⁻² (low heat) or 50 kWm⁻² (high heat), and subjected to an air flow entering from below at rates of 50 lpm (low flow) or 200 lpm (high flow). This configuration is shown in Figure 2, adapted from Haslett et al., (2017).

The fuel used in Edinburgh was Hevea wood from Côte d'Ivoire. The plume was collected in a hood before entering the exhaust
tube. Air samples were collected simultaneously at two points: (A) in the exhaust tube, where CO and $CO_2$ were measured directly and a further sample was diluted in pure nitrogen by a factor of 100 for the online measurement of BC and OC concentrations. An Aerosol Mass Spectrometer (AMS) was used to measure the concentration of organic aerosols and other non-refractory species and a Single Particle Soot Photometer (SP2) to measure refractory black carbon (BC) mass



concentration. The $CO_2$, CO and $O_2$ concentrations in the plume were recorded at a frequency of 1 Hz at the exhaust tube by the FPA.

(B) At the exit of the exhaust pipe for filter sampling (offline measurement). A QTRAK analyser was used to continuously measure CO and $CO_2$ concentrations. Two aerodynamic sampling lines (PM2.5 and TPM) were used with pumps, counter,
cut-off heads and filter holder with 47 mm quartz filters.

Eight combustion tests were carried out during this experiment. Two of these were made with infrared lamps set at 30 kWm$^{-2}$ and incoming airflow of 200 lpm (low heat and high flow: hFl), 3 tests at 50 kWm$^{-2}$ and 200 lpm (high heat and high flow: HFl) and 3 other tests at 50 kWm$^{-2}$ and 50 lpm (high heat and low flow: Hfl).

### 3 Sample analysis

Two types of sampling were performed during our EF measurement experiments: collection of particles on Whatman quartz filters and Speciated NMVOCs (Alkanes, Alkene, Carbonyl and Aromatic Compounds) on absorbent tubes. In order to capture a large spectrum of VOCs two types of sorbent tube were used: Tenax TA 60-80 meshes (250 mg, 2,6-diphenyl-p-phenylene oxide), and multi-sorbent cartridges composed of Carbopack C (200 mg) and Carbopack B (200 mg) 60-80 mesh (graphitized carbon black). The analysis of Tenax tubes was performed at the *Laboratoire* de Météorologie Physique (LaMP), using a gas
chromatograph - mass spectrometer system (GC/MS, Turbomass Clarus 600, Perkin Elmer) connected to an automatic thermal desorption unit (Turbomatrix ATD). Each cartridge was desorbed at 270 °C for 15 min at a flow rate of 40 mL/min, reconcentrated on a second trap, at -10°C containing Tenax TA. After the cryofocussing, the trap was rapidly heated to 300°C (40°/s) and the target compounds were flushed into the GC. The separating column used was a capillary PE-5MS (60m×0.25mm×0.25μm, 5% phenyl – 95% PDMS, Perkin Elmer) and the GC temperature profile was ramped from 35°C for
5 min, heating at 8°C min$^{-1}$ to 250°C and hold for 2 min.  The mass spectrometer was operated in a Total Ion Current (TIC) from 35 to 350 m/z amu, and all chromatography parameters were optimized to enable the separation of 16 compounds from C5-C10 NMVOCs. Calibration was performed by analyzing conditioned cartridges doped with known masses of each compound, presented in standard low-ppb level gas solutions (purchased from the National Physical Laboratory, UK). The cartridges were then analyzed with the aforementioned method and calibration curves were obtained for each compound.
Carbopack cartridges were analyzed at *SAGE Department (IMT Lille Douai)* with an analytical system ATD-GC-FID, already described in (Detournay et al., 2011; Ait-Helal et al., 2014). This method allowed the separation and identification of more than 50 compounds, from C5-C16 NMVOCs, including carbonyls, ketones, terpenes and intermediate VOCs (C11-C16).

Gravimetric analysis of quartz filters (providing TPM, PM10, PM2.5, etc,) was performed by comparing the difference in weight of the filters before and after exposure. Weighing was performed using a SARTORIUS Microbalance with 1.95 μg
sensitivity. After the gravimetric analysis, the laboratory two-step thermal method from Cachier et al., (1989) was applied for the separation and the analysis of black and organic carbon aerosol contents (BC and OC, respectively). Note that the detection stage was adapted since in our instrument (G4 ICARUS), aerosol carbon content is quantified from $CO_2$ by a non-dispersive





infrared (NDIR) detector, instead of coulometry. The relevance of the use of thermal method was validated by comparing results of 10 samples analysed by the thermo-optical method (IMPROVE method, Chow et al., 1993, 2001).We performed a linear regression analysis of all values obtained for both methods for TC, BC and OC. The analysis of the regression coefficients (given here in terms of $R^2$) show that suitable correlations were found among the thermal and thermo-optical

methods for TC, BC and OC values. The thermal method gives 94% of OC and 90% of the BC measured with the thermo-optical method. Examples of regression plots are given in Supporting Information Figure S1 for the thermal methods against the thermo-optical method. After this comparative study of analyses by these two methods, the thermal method was used for the subsequent analyses. Two similar aliquots of the same filter were then separately analysed. One portion was directly analysed for its total carbon content (TC). The other portion was firstly submitted to a pre-combustion step (2 h at 340°C under

pure oxygen) in order to eliminate OC, and then analysed for its BC content. Organic carbon (OC) concentrations were calculated as the difference between TC and BC.

## 4 Results and Discussions

### 4.1 Ground field measurements

Field measurements allowed mean values EFs for residential sources using charcoal (CH) and fuel wood burning (FW) to be obtained; for charcoal making (CHM); for open waste burning fires (WB) and for vehicle traffic (car, bus, truck, light duty vehicles, two-wheeled two-stroke and four-stroke vehicles) by energy source (Diesel and Gasoline) and by age group (new and old). As aforementioned, the studied species are CO, $CO_2$, carbonaceous particles in particular (BC, OC and TC), speciated NMVOCs compounds (C5-C16 alkanes, C5-C11 carbonyls, C4-ketones C6-C9 aromatics and 13 species of terpenes) and total

aerosol mass (TPM). Speciated NMVOCs list is shown in Table 7. At least three tests were performed for each studied sources to reflect the reproducibility of results. Mean EF values are obtained by the arithmetic method for each sources or geometric method for each sector.

### 4.1.1 Combustion characteristics of the studied sources

The mean MCE value for all ground-studied sources are shown in Figure 3.

It can be seen that Fossil fuel (FF) sources (DL, MO) have MCEs in the range between 0.9-1, aside from two wheeled two stroke engines (TW 2T), which had an MCE of 0.65. As expected, TW 2T with a mixture of gasoline and oil, presents an incomplete combustion with more abundant smoldering products. MCE of biofuels are found between 0.6 and 0.9, with the highest values for Iroko wood, which burns with more flaming than other biofuel tested in this study.




Table 1 summarizes Modified Combustion Efficiency (MCE) and $\Delta CO/\Delta CO_2$ values for all sources studied. As already recalled, the $\Delta CO/\Delta CO_2$ ratio is a combustion quality indicator: the smaller this ratio, the better the combustion. These ratios ranged between 0.006 and 0.32, showing the importance of the range of pyrolysis in the carbonization process and the various smoldering conditions depending on the mentioned experiments.

Note that charcoal making (CHM) $\Delta CO/\Delta CO_2$ (value) ratios are higher than those of wood burning ($0.09 \pm 0.03$ versus $0.17 \pm 0.04$), as wood burning generally occurs more in the flaming phase. Also, $\Delta CO/\Delta CO_2$ ratio is higher in CHM than in charcoal burning ($0.219 \pm 0.098$). Diesel (DL) and motor gasoline (MO) had the smallest $\Delta CO/\Delta CO_2$ ratio for all studied sources; this shown that fossil fuel combustion occurs generally more in the flaming phase. Also, the comparison between CHM MCE and $\Delta CO/\Delta CO2$ value for this study and published values show that these values are very close. Indeed, our study presents MCE

values of 0.76 whereas Bertschi et al., (2003), Lacaux et al., (1994) and Smith et al., (1999) respectively obtained 0.78, 0.81 and 0.77. In terms of $\Delta CO/\Delta CO_2$ ratio, 0.32 is here obtained whereas Cachier et al., (1996), Bertschi et al., (2003), Lacaux et al., (1994) and Smith et al., (1999) respectively had 0.26, 0.28, 0.24 and 0.30.

Finally, note that composition of combustion aerosol is also impacted by the combustion characteristics. Indeed, as shown by Figure 4, the more the $\Delta CO/\Delta CO_2$ ratio increases, the more the BC/TC ratio decreases. This reveals that combustion under

smoldering conditions such as TW, CHM and wood combustion (pink and red points, respectively) produce relatively more important organic carbon concentrations than more complete combustions such as diesel, gasoline and WB combustion (green, blue and brown points). This is in agreement with previous works (Andreae and Merlet, 2001). Such variations have also an impact on EFs values, which are now presented source by source.

### 4.1.2 EF of residential sources

The EF for fuelwood sources were calculated following relationship (1), and using fc = 46% (Brocard et al., 1996) and MCE of $0.76 \pm 0.11$ and $0.92 \pm 0.02$ for Hevea and Iroko woods respectively (Fig 3). Results of EF(OC) values given in table 1 are $13.11 \pm 5.41$ and $5.46 \pm 1.66$ g/kg dry matter (dm) for Hevea and Iroko wood respectively, while for BC, EF were found to be $1.22 \pm 0.52$ and $0.43 \pm 0.33$ g/kg dm respectively. Hevea wood emitted more carbonaceous particles than Iroko wood. Assuming that these woods are the primary wood fuel sources in the tropical African region, calculations suggest that on

average, each wood type contributes 50% to the total wood burned. Following this assumption, a value of $0.82 \pm 0.39$ g/kg dm for BC and $9.29 \pm 3.82$ g/kg dm for OC are representative of the EFs typical for these areas. Such values are in agreement with those obtained by Radke et al., (1991), for OC and those of Brocard et al., (1996) for BC (see table 2). Our measured EFs are generally higher than values found in literature. These differences may be explained by the wide variety of wood used in the different studies as well as the different measurement methods. Such a difference is enhanced when studying the $\Delta CO/\Delta CO_2$

ratio. However, our values are in agreement with to those of McDonald et al., (2000); Oros and Simoneit, (2001), who burned same kind of woods (hardwood), using same methodology. CH EFs were obtained by averaging EFs of several tests. fc of 71.5% (Brocard et al., 1996) and MCE of $0.83 \pm 0.06$ were used for such calculations. Table 2 presents the mean EF for charcoal burning with literature data.





It may be seen here, our results for charcoal burning EFs are comparable to the range of data given by literature.

The carbon balance method cannot be used directly to calculate charcoal making (CHM) EF (Bertschi et al., 2003), as during CHM, part of the carbon which is emitted by wood burning is found in charcoal, ash and in the pyroligneous liquid. Then, less than 50% is emitted into the atmosphere. As shown previously, Lacaux et al., 1994, Cachier et al., 1996, Smith et al., 1999 and Bertschi et al., 2003 estimate the fraction of carbon emitted as atmospheric gases to be 35%, 35%, 37%, and 45 % respectively. Moreover, Cachier et al., (1996) estimate that during CHM if 35% of the carbon content of wood is emitted into the atmosphere, 89% is emitted as CO and $CO_2$ during the smoldering phase. Since the CHM conditions described in Cachier et al., (1996) are similar to those of this study (pure smoldering because of an average MCE of 76% (figure 3) (Akagi et al., 2011)), we consider 35% of the carbon content of wood to be emitted into the atmosphere and 89% of this carbon to be emitted as CO and $CO_2$ for CHM. Thus, in order to obtain an EF in g/kg dm, the EF in g/kg of carbon was multiplied by 0.35 and 0.89 respectively. As shown in table 1, CHM EFs of this study are comparable to those of Brocard, 1996 and Cachier et al., (1996) for OC and are a factor 2 lower for BC. This difference could be explained by a wide variety of wood used for CHM (e.g. wood type or wood moisture) but also by the combustion conditions in furnaces.

### 4.1.3 EFs for road traffic sources

EFs for road traffic sources were calculated following relationship (1). The fraction of carbon contained in diesel (IPCC, 2006; Kirchstetter et al., 1999) and in gasoline (IPCC, 2006; Ban-Weiss et al., 2010) was assumed to be 85% in order to obtain an EF in g/kg of fuel burned. Table 3 summarizes BC, OC and TPM EFs from our measurements and literature by age group (recent and old), energy or fuel type (gasoline and diesel) and measurement methods. Results show that our measured EF(BC) (0.0012±0.0006) for recent light-duty gasoline vehicles (LDGV) are within the range of literature values, while for aged vehicles they are 100 times higher. This high factor for old vehicles can be explained by the fact that literature EFs are mostly measured in developed countries where vehicle emission regulations exist (catalytic converter and diesel particulate filters, e.g. standard EURO 6). In Africa, these types of regulation are rare. Measured EF(BC) are higher than EF(OC) for diesel, which is coherent with Fig.4 and is in agreement with Pant and Harrison, (2013); Chiang et al., (2012); and Grieshop et al., (2006). Whereas, measured EF(BC) are lower than EF(OC) for gasoline as previously shown by some studies such as Pant and Harrison, (2013); Ntziachristos et al., (2007). Table 3 also shows that diesel vehicles emit more particles (TPM) than gasoline.

It is important to note that, as mentioned earlier, the differences observed between our values and others can be also due to the method by which EFs were established. For example, EFs (BC) obtained from roadside measurement methods (including all type of vehicle) are globally higher than EFs from direct exhaust pipe measurements, which is the method applied in this study.

To calculate the mean EFs for light-duty and heavy-duty vehicles (Table 3), we assume that the park vehicle is constituted by 60% of old vehicles and 40% of recent vehicles. Then mean EFs (g/kg fuel), of BC, OC and TPM for light-duty diesel vehicles (LDDV) including private cars and taxi are respectively 3.35 ± 2.20, 2.03 ± 1.13 and 35.82 ± 21.40. They are higher than EFs for heavy-duty diesel vehicles (HDDV) including truck and bus, respectively of the order of 2.20 ± 1.05, 2.50 ± 1.43 and 31.00



± 15.80. Regarding gasoline vehicles, EF measurements were carried out for LDGV, which constitute the majority of the fleet using gasoline. Mean BC, OC and TPM EFs are 0.62 ± 0.49 g/kg fuel, 1.10 ± 0.77 g/kg fuel and 7.0 ± 2.80 g/kg fuel respectively for gasoline which are lower than those of heavy-duty diesel vehicle (HDDV) as shown by Robert et al., (2007). Table 4 summarizes the mean EFs results measured during this study for road vehicles by fuel type (gasoline and diesel) compared to

those from previous studies. Calculations were based on the vehicle proportion given by Direction Generale des Transports Terrestres (DGTT) of Côte d'Ivoire. In addition to the previous assumption (60% of total vehicles were considered old and 40% new models), it was assumed that 77% of total vehicles are light duty vehicles and 23% heavy duty vehicles. Finally we compared OC/BC ratio of this study to that of COPERT (Ntziachristos et al., 2009) for LDGV, LDDV and HDDV. LDDV OC/BC ratio for our study (47-63 %) (Table 4 ) are in the COPERT OC/BC ratio range 40-70 % typical for Euro1 –

Conventional norm while HDDV OC/BC ratio (205%) are quite similar to those COPERT EURO 4 HDDV ratio (300%): this shows that most African HDDV and LDDV are relatively consistent with EURO4 and EURO1 standards, respectively. For LDGV, OC/BC ratio is relatively close to COPERT LDGV norm EURO2 – EURO4 ratio range (250 – 300%). We have also compared our mean EFs values for road traffic (diesel and gasoline) with EFs value used in the most recent African emission inventory (Liousse et al., 2014). Our EF values are higher than their values for gasoline (4 times higher for BC and 2 time

higher for OC).Our EFs values are slightly lower for diesel.

EFs of two-wheeled (TW) vehicles were classified according to age (old and new), and engine type. We distinguished the two-stroke engines using a mixture of gasoline and oil from the four-stroke engines. For recent TW two-stroke engines, BC and OC EFs are 2.26 ± 1.40 g/kg and 26.0 ± 1.10 g/kg respectively, while, 0.11 ± 0.01g/kg and 0.45 ± 0.13 g/kg are found for new TW four-stroke engines (Table 5). The same difference is observed between old TW four-strokes and old two-strokes, with

the exception of EFs (BC). This implies that TW two-stroke engines emitted much more OC particles (OC/BC of 36) than TW four-stroke engines (OC/BC of 7). Such values for both BC and OC observations from two-stroke engines can be explained by incomplete combustion due to gasoline-oil mixtures used in these engines. This has already been highlighted by Volckens et al.,(2008) for particulate emissions when studying two-strokes engines. In addition, the old/new ratio of BC(EF) for TW- 2 strokes is around 1.5. While the same ratio for OC(EF) is around 5, which is 3 times greater than that of BC(EF). Similarly,

for TW-4 strokes, the old/new ratio of EF(OC) is a factor of 2 greater than the ratio of EF(BC). That shows that OC emissions are more enhanced (doubled or tripled) in older TW vehicles compared with those of BC.

The global mean EF of TW vehicles was calculated following Assamoi and Liousse, (2010) using the assumptions that 40% are two-stroke and 60% four-stroke engines, and that 40% are new and 60% old vehicles. In that context, the mean EF values (Table 5) are in agreement with values found by Assamoi and Liousse, (2010).

It is interesting to note that EFs of carbonaceous particles for two-stroke motorcycles in Africa are higher than those in Europe (see Table 5). These higher EFs may be explained by the fact that African TW two-stroke vehicles are often older and second hand (used) vehicles imported from Europe. In addition, the fuel quality used in Africa is bad (UNEP, 2016). Indeed, Assamoi and Liousse, (2010) showed that a large part of the fuel used by two-wheeled vehicles in West Africa is adulterated and smuggled and therefore of poor quality compared to European standards. The OC/BC ratio for different experiments has been



evaluated and summarized in Table 5. This ratio varies between 4.1-7 for TW four-stroke and between 11.4-36 for TW two-stroke, due to the bad combustion in TW two-stroke engine. Such values are in agreement with OC/BC ratio given by Bond et al., (2004) and Guillaume and Liousse, (2009) for TW two-stroke. Note that, the OC/BC ratio given by COPERT TW are ranged between 2.5 – 9 (corresponding to Euro 3 - conventional norms) (Ntziachristos et al., 2009) which is much lower than
our values and rather comparable with TW four stroke engine.

### 4.1.4 EFs for open solid waste burning

EFs of pollutants for open solid waste burning were calculated using Equation 1 adapted for this source. In that case, Modified Combustion Efficiency (MCE) was replaced by carbon oxidation factor (COF) defined as the ratio between the amount of burned carbon and the amount of carbon initially present in the sample. Carbon content of household waste of 46% was used
(Lundin et al., 2013 ; Wiedinmyer et al., 2014) with a COF of 58% (Fiedler et al., 2010). The averaged EFs for open solid waste burning are presented in Table 6 with associated standard deviation. During our measurements, several phases of combustion were observed: a flaming phase was observed during dry waste combustion, and a smoldering phase during wet waste combustion in landfills. These various fire types and the waste composition explain the relatively high value of the associated standard deviation. EF (g/kg-solid waste) of BC, OC and TPM, are 2.80 ± 3.30, 6.44 ± 4.60 and 87.90 ± 32.90,
respectively. As expected, waste burning emits more OC than BC. In addition, the relative high value of TPM suggests that the existence of other kinds of particulate matter (such as ions or metals) is also emitted during waste burning. When comparing values found during this study to those of Christian et al., (2010) which deals with carbonaceous particles EFs, it is noted that EF(OC) are of the same order of magnitude, while the EF(BC) of our work is higher than that of Christian et al., (2010) with a factor 4 of difference. This high EF(BC) found in the present work may be explained by differences in the solid waste
composition from Côte d'Ivoire and Mexico, where the measurements of Christian et al., (2010) were carried out. Moreover, it can be also explained by the fact that more flaming was observed during our measurements.

### 4.1.5 NMVOCs EF

Fifteen common NMVOC species ($C_5$ to $C_{10}$) were identified and quantified from sorbent tube measurements and are reported in Table 7. Globally, the dominant NMVOC species emitted during our EF measurements include toluene, *m+p*-xylene, 1,2,4-
trimethylbenzene (124-TMB), ethylbenzene, *o*-xylene, 1,3,5-trimethylbenzene (135-TMB) and heptane. Most of these compounds are important species in terms of atmospheric reactivity, generally involved in photochemistry processing and in the formation of secondary pollutants like ozone (Pandis and Seinfeld, 2006). Aromatic compounds also have high secondary organic aerosol potentials (Derwent et al., 2010), so contribute to the formation of these particles. While they are usually associated to biogenic emissions (Kesselmeier and Staudt, 1999), isoprene and terpenes (limonene, α-pinene and β-pinene)
were also observed in the EFs of almost all anthropogenic sources. Table 7 presents EF values of such species for all the studied sources. It is important to note that standard deviation values are high, reflecting the range of uncertainties linked to the two set of analysis (see methodology section) and also to the different sources associated to the emission sector analysed.



As it may be seen, road traffic NMVOC EFs are the most important, especially for two-wheeled two-stroke engine (TW 2T). The EFs for TW are up to three orders of magnitude greater than those observed for diesel vehicles (heavy-duty diesel vehicles: HDDV; light duty diesel vehicles: LDDV) and light duty gasoline vehicles (LDGV). They are dominated by alkanes and aromatic compounds (Tsai et al., 2003). Likewise, we highlight the presence of isoprene and terpene emissions in TW sources,

whose contribution cannot be neglected. In terms of engine differences, 4-stroke engine emissions had lower EFs than those observed for 2-stroke engines. This result is in agreement with other works (Tsai et al., 2000, Montero et al., 2010), which analysed the concentration of individual VOC in the tailpipe exhaust showing the differences between 2-stroke and 4-stroke engine emissions.

Gasoline vehicle NMVOC EFs are higher than diesel vehicles and dominated by aromatics compounds, including xylenes

(45%), trimethylbenzenes (25%) and toluene (15%). The main differences associated with both fuel emission profiles were related to the higher relative contribution of benzene (37% for LDDV) and alkanes (18% for HDDV) for diesel vehicles. Charcoal making (CHM), wood (FW) and charcoal burning (CH) emissions were characterized by the abundant presence of alkanes, benzene, and xylenes. Particularly in the case of FW, important EF of terpenes (15%) and isoprene (13%) contributions were observed. The most abundant NMVOC species observed for open waste burning (WB) were terpenes (38%)

followed by toluene, trimethylbenzenes, benzene and alkanes respectively. The sum of $C_5$-$C_{10}$ NMVOC for this study show a similar EF range in comparison with the sum of non-methane organic compounds found in the literature (Christian et al., 2010). The variability in the WB emission factors measured are important (up to 100%), and further analysis should be performed in order to find the cause of this high standard deviation.

Individual NMVOC species were aggregated into species groups as proposed by the GEIA initiative and introduced in the last EDGAR VOC inventory (Huang et al., 2017) (Tables S1, S2 and S3). In this way, a larger VOC database was considered, including 13 species of terpenes, intermediated VOCs (iVOCs from C11-C16 $n$-alkanes), ketones and carbonyls compounds for a reduced number of sources (Table S3). The main differences obtained from this exhaustive speciation were related to the contribution of alkanes (VOC6 50%, being the iVOCs the most important fraction) and aldehydes (VOC22, 13%) for HDDV

sources. The contribution of heavy alkanes from diesel was also observed in other studies (Ait-Helal et al., 2014; Dunmore et al., 2015). In the same way aldehydes presented a considerable contribution in CH and CHM emissions and terpenes were also significant (14%) in wood burning emissions (FW).

The determined EF for gasoline (LDGV) and diesel (LDDV and HDDV) vehicles and wood burning have been compared (Figure 5) to the ones from the literature (McDonald et al., 2000; Gentner et al., 2013; Evtyugina et al., 2014) in order to

evaluate the magnitude of the West African anthropogenic emissions of VOCs. Numbers are reported in table S4 of the supplement material. Regardless of the motorization, the EFs in West Africa are higher than the most recent ones reported in California in the Cadelcott tunnel by Gentner et al., (2013). The differences span two orders of magnitude. The EFs for wood burning are also higher by between a factor of 2 and a factor of 100 than those reported in the literature for different types of hardwood used in woodstoves and fireplaces (Figure 5). The presence of isoprene and monoterpenes in WB emissions is





consistent with the literature and is as significant as the presence of aromatics. This comparison highlights the strong emissions of anthropogenic VOCs and the usefulness of in-situ measurements in West Africa region to derive realistic emission factors.

### 4.2 Combustion chamber measurements for fuelwood burning

Combustion chamber EFs were calculated using the carbon balance method in Eq1. In order to compare these values to AMS and SP2 measurements, the following equation was used:

$$EF = \frac{([C_x]_{smoke} - [C_x]_{backgroung}) * Q_{chamber} * t}{m_{burned}} \qquad (Eq3);$$

In Eq3, EF (x) is the emission factor of the pollutant x in g/kg fuel burnt; $[C_x]_{smoke}$ and $[C_x]_{background}$, the concentration of x in the fresh smoke plume and the background air, respectively, $Q_{chamber}$, the air flow entering the chamber, t, the sampling time

and $m_{burned}$, the mass of burnt fuel.

Table 8 summarizes EFs of BC, OC and TPM carried out in the combustion chambers. As it may be seen, field EF values obtained with the same methodology "filter sampling" are a factor of 16 and 11 higher than those of Lannemezan combustion chamber for BC and OC respectively, and a factor of 16 and 28 higher than those of Edinburgh combustion chamber. This important difference between field and combustion chamber results may be linked to the high dilution of plumes occurring in

combustion chambers. Indeed, field EF measurements allow plume dispersion to be avoided. In terms of quantities, ground field values are representative of primary emissions, which have to be computed in emission inventories algorithms. However, it may be underlined that, combustion chamber measurements are very complementary to ground field measurement. First, it allows more measurements to be performed than on the ground. Second, it allows a better understanding of the impact of combustion processes on aerosol composition and size class (e.g. PM2.5, PM1).

### 4.2.1 EF comparison between field and combustion chamber measurements for wood burning

As noticed earlier, ground field EFs are higher than Lannemezan combustion chamber EFs. Quantitavely, it may be seen that a dilution factor of around 8 exists between field and combustion chamber measurements. Indeed $CO/CO_2$ field measurements ($\approx 0.32$) are 8 times higher than that of Lannemezan ($\approx 0.04$), and $CO_2$ is roughly the same. The same factor is obtained for

the $EF_{field}/EF_{Lannemezan}$ ratio for BC and total mass. Moreover, both BC/TC (Total Carbon) ratios are very similar. That means that FW combustion at Lannemezan qualitatively mimics field FW combustion well. This is not so clear with the Edinburgh results. Indeed, Edinburgh EF results show that there is more flaming during the first two tests than during the field tests due to higher BC/TC ratio ($\approx 13$) rather than 8.9 (in the field). Finally, we have compared EFs (BC and OC) obtained at Edinburgh combustion chamber with two sampling methods: using the filter and the AMS. The differences are more pronounced with

BC/TC ratio obtained with AMS-SP2 of the order of 0.35 which shows a strong flaming condition. In that context, EF(BC) obtained from AMS-SP2 measurements are 100 times higher than those obtained from filter methodology.



### 4.2.2 Wood burning EFs per size-class

Relative contributions of different size classes (PM10, PM2.5, PM1, and PM0.25) to total size of BC and OC EFs are shown in Table 9. As it may be seen, such a contribution is less variable for BC EFs from fine particles (PM2.5: 81%) to ultrafine particles (PM0.25: 77%) than for OC EFs which varies from 72% to 51% for PM2.5 and PM0.25, respectively. This means that BC EFs particularly predominate in the ultrafine size fraction. This results is in line with that reported by previous works on fuel wood burning (Guofeng et al., 2012; Danielsen et al., 2011; Purvis et al., 2000). The same results was found by Lu et al., (2012) with BC and OC emissions from a diesel engine. Table 9 also shows that BC/TC is consequently different in the different aerosol size fraction with bigger values in the ultrafine sizes (0.094) than in the coarse ones (0.064). The domination of fine particles from fuelwood burning is a health concern for those who use wood for cooking, since fine particles can penetrate deeper into the lungs and are often associated with many toxic compounds (Englert, 2004; Pope et al., 2009; Val et al., 2013).

### 5. Conclusion

This study characterized the emissions of many sources specific to Africa. EFs of BC, OC, TPM and specified NMVOCs were determined for biofuel (tropical fuelwood, charcoal and charcoal making), fossil fuels used in traffic (gasoline and diesel) and solid waste burning. Ground field EF measurements were performed for all studied sources as well as in combustion chambers for fuelwood in order to obtain EFs per size fractions.

During field measurements, several tests were performed in order to gain the more representative EFs of the studied sources. The mean EF of BC, OC and TPM are $0.83 \pm 0.39$, $9.29 \pm 3.82$, $34.54 \pm 20.6$ g/kg dry matter (dm) for wood burning and $0.65 \pm 0.30$, $1.80 \pm 2.80$, $12.75 \pm 9.0$ g/kg dm for charcoal burning, respectively. In general, EFs for biofuel burning are comparable to the range of those found the literature, wood burning emitting more particle than charcoal burning.

EFs for fossil fuel burning in traffic are strongly dependent on vehicle age and maintenance. More aged and more poorly maintained vehicles produced higher EF values and were up to 100 time higher than literature EF values. These older vehicles are the most used in African countries and can be characterized as typical "African EFs". In contrast, EFs for new vehicle models are comparable to published EFs values. The mean EF of BC, OC and TPM are $0.62 \pm 0.49$, $1.10 \pm 0.77$, $7.0 \pm 2.80$ g/kg fuel for road gasoline and $3.10 \pm 1.96$, $2.14 \pm 1.20$, $34.70 \pm 20.1$ g/kg fuel for road diesel, respectively.

Moreover, the EFs of more than 50 NMVOC have been determined for the studied sources for the first time in West Africa. These EFs showed that emission profiles are dominated by aromatics (up to 80% for some traffic-related sources). The greatest emissions are observed for two-stroke two-wheelers which can be higher than three orders of magnitude compared to the EFs observed from LD and HD vehicles. The presence of terpenes in biofuel burning emissions was considerable as well as heavy alkanes (iVOCs), reaching up to 50% in diesel related sources. Comparison to recent literature worldwide points out the greatest levels of anthropogenic African EF for NMVOC and the relevance of in-situ measurements to derive realistic and representative emission factors.



In the combustion chamber measurements, EFs were determined by size class showing that BC is mainly in the fine fraction. The significant difference between the combustion chamber and field measurements suggests that EFs and chemical composition are strongly affected by variables that differ according to field and laboratory procedures. Even if ground field values are more representative of primary emissions, combustion chamber allows to perform more measurements than in the field, allowing to better understand the impact of combustion processes on aerosol composition and size class. Nevertheless, these two types of measures remain complementary. EFs data obtained in this work are specific and representative of African combustion sources. This unique database will be useful for improving and updating African emission inventories allowing better assessments of climatic, air quality and health impacts. This study may also help decision makers to set up new politics on energy sources and particularly traffic, domestic cooking as well as waste burning sources.



**Acknowledgment**

This work has received funding from the European Union 7th Framework Programme (FP7/2007-2013) under Grant agreement no. 603502 (EU project DACCIWA: Dynamics-aerosol-chemistry-cloud interactions in West Africa). The author would like to thank the AUF (Agence Universitaire de la Francophonie) for funding his stays in Laboratoire d'Aérologie of Toulouse (3

5    months per year during 3 years). Part of the analysis of NMVOC tubes were graciously performed at the SAGE Department at IMT Lille Douai (France) by Thierry Leonardis and Stéphane Sauvage.



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





**Lists of Figures**



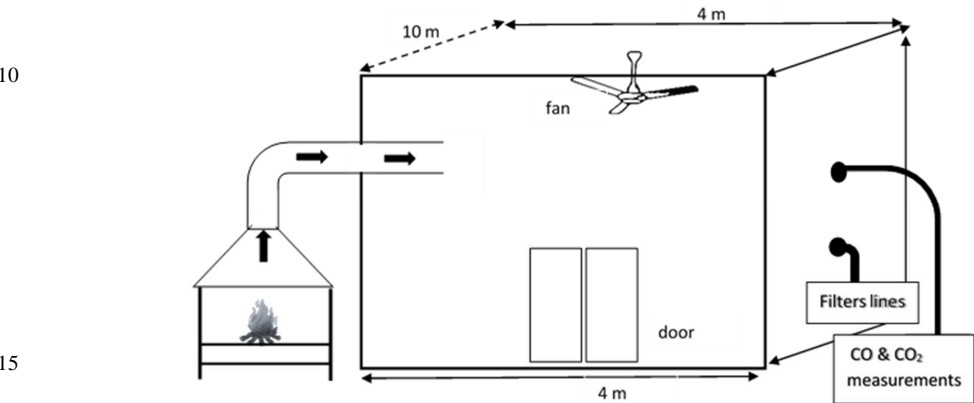

**Figure 1: Schematic view of Lannemezan combustion chamber**





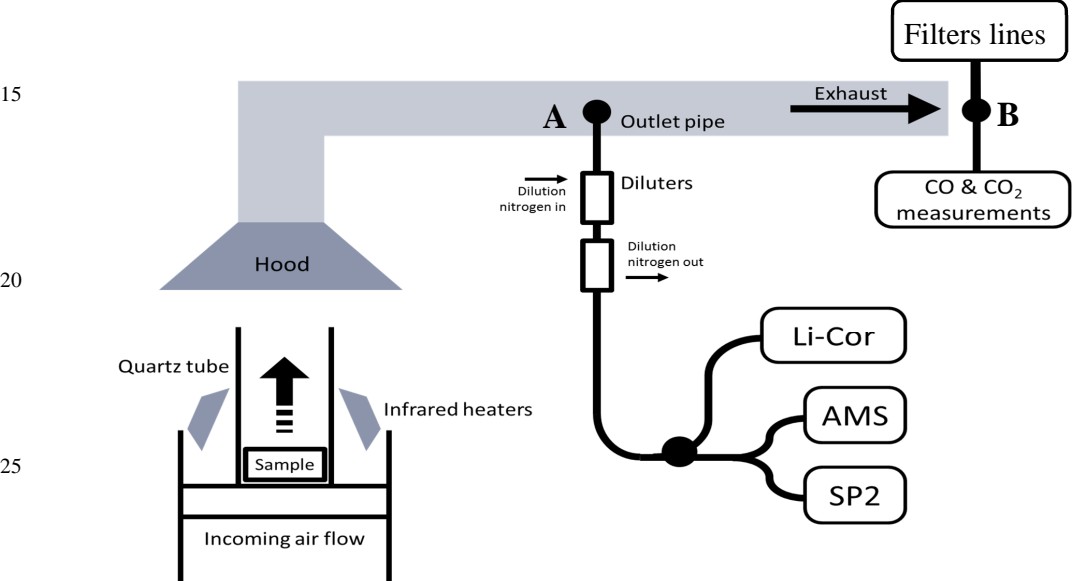

**Figure 2: Schematic view of the FM-global fire propagation apparatus with a diagram of the gas and particle sampling system (adapt from Haslett et al., (2017)**





**Figure 3: Modified Combustion Efficiency of different sources studied (charcoal burning:CH, charcoal making:CHM, diesel: DL, wood burning:FW, gasoline:MO, two-wheeled vehicles: TW and waste burning:WB)**



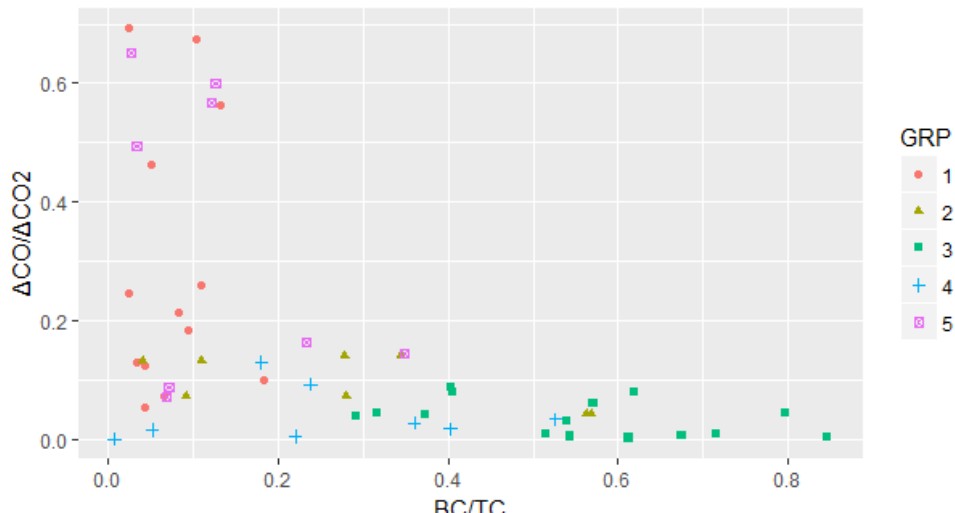

**Figure 4: Covariation of the $\Delta CO/\Delta CO_2$ and BC/TC (Total Carbon) ratios for all studied sources (1-wood, 2-Waste Burning, 3 Diesel, 4-Gasoline, 5-Two-Wheeled vehicles)**


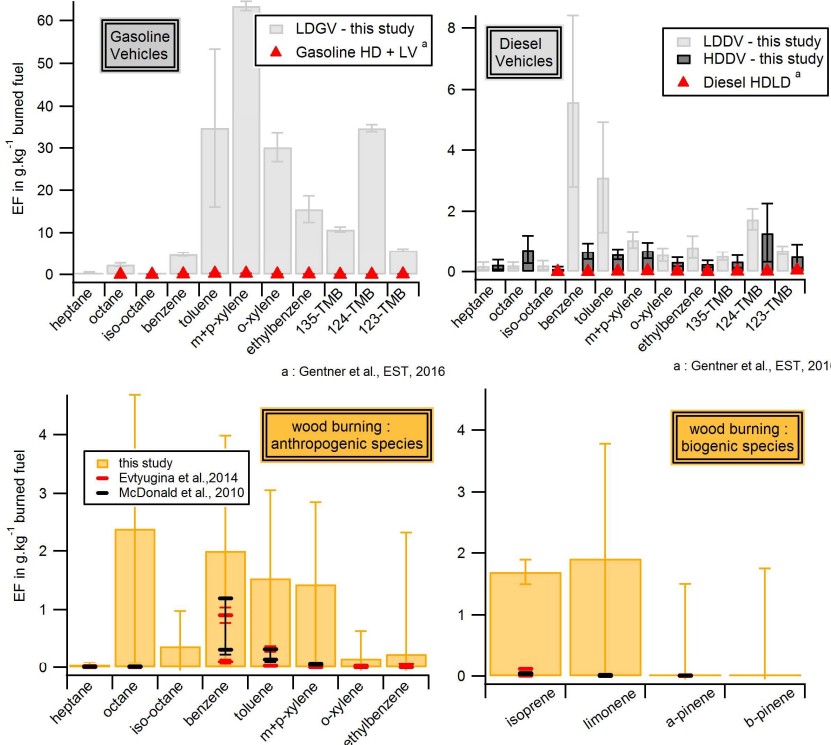

**Figure 5: Comparison of NMVOCs EF with the literature for traffic (light duty gasoline vehicles (LDGV) and light duty diesel vehicles (LDDV) and heavy-duty diesel vehicles (HDDV)) and wood burning. The bars correspond to the standard deviations of measurements**



**Lists of Tables**



**Table 1: Modified Combustion Efficiency (MCE) and ΔCO/ΔCO₂ ratio for all the studied sources**

| Emission source | MCE | $\Delta CO/\Delta CO_2$ |
|---|---|---|
| Diesel | 0.94-0.99 | 0.01-0.06 |
| Gasoline | 0.94-0.99 | 0.01-0.06 |
| TW | 0.65-0.89 | 0.11-0.65 |
| Wood | 0.92-0.76 | 0.09-0.17 |
| Waste burning | 1.00±0.00 | 0.09±0.04 |
| Charcoal | 0.83±0.06 | 0.23±0.09 |
| CHarcoal Making | 0.76±0.05 | 0.32±0.10 |



**Table 2: EFs of residential sources for this study and those from literature**

| Emission sources | Type | Reference | EF (g/kg dm) | | | OC/BC |
|---|---|---|---|---|---|---|
| | | | BC | OC | TPM | |
| Wood | Iroko | This work | 0.43 ± 0.33 | 5.46 ± 1.66 | 13.86 ± 2.17 | 12.8 |
| | Hevea | | 1.22 ± 0.52 | 13.11 ± 5.41 | 55.22 ± 25.20 | 10.7 |
| | Mean | | 0.83 ± 0.39 | 9.29 ± 3.82 | 34.54 ± 20.67 | 11.23 |
| | | Radke et al., 1991 | 2.52 | 8.41 | nd | 3.3 |
| | | | 4.21 | 14.04 | nd | 3.3 |
| | fireplace | McDonald et al., 2000 | 0.40 | 3.49 | 5.95 | 8.8 |
| | woodstove | | 0.31 | 2.86 | 4.02 | 9.2 |
| | | Brocard et al., 1996 | 0.55 ± 0.3 | 5.0 ± 3.6 | nd | 9.1 |
| | various | Oros et al., 2001b | 0.145-1.85 | 2.05-25.48 | nd | 9-43 |
| Charcoal | | This work | 0.65 ± 0.30 | 1.78 ± 2.80 | 12.75 ± 9.03 | 2.7 |
| | | Brocard, 1996 | 0.2 | 2 | nd | 10 |
| | | Liousse et al., 2014 | 0.75 | 2.3 | nd | 3 |
| | | Roden and Bond, 2006 | 0.2 | 1.5 | nd | 7.5 |
| Charcoal making | | This work | 0.20 ± 0.14 | 3.50 ± 0.90 | 35.71 ± 19.81 | 17.5 |
| | | Cachier et al., 1996 | 0.46 ± 0.1 | 4.4 ± 0.6 | nd | 9.56 |
| | | Brocard, 1996 | 0.4 | 3.6 | nd | 9 |




5  **Table 3: Emission factors for gasoline and diesel vehicles by age group and those in the literature, from different measurement methods.**

| Fuel | Age | EFs (g/kg-fuel) | | | Study | Method |
|------|-----|------|------|------|-------|--------|
| | | BC | OC | TPM | | |
| Gasoline | New car | 0.001 ± 0.001 | 0.042 ± 0.04 | 3.02 ± 0.3 | This work | Measurement around tailpipe |
| | Old Car | 1.03 ± 0.83 | 1.80 ± 1.26 | 9.63 ± 4.44 | | |
| | LDGV | 0.62 ± 0.49 | 1.10 ± 0.77 | 7.00 ± 2.80 | | |
| | | 0.002 | nd | nd | Wang et al., 2016 | Measurement in tailpipe |
| | | 0.026 | nd | nd | Ban-Weiss et al., 2008 | Roadside measurement |
| | | 0.152 | nd | nd | Liggio et al., 2012 | Roadside measurement |
| | | 0.002 – 0.01 | nd | nd | Forestieri et al., 2013 | Chasis dynamometer measurements |
| Diesel | Recent car | 1.26 ± 0.66 | 0.60 ± 0.25 | 9.13 ± 4.08 | This work | Measurement around tailpipe |
| | Old Car | 4.74 ± 3.2 | 2.97 ± 1.71 | 53.62 ± 33.0 | | |
| | Recent bus | 0.35 ± 0.01 | 0.72 ± 0.15 | 6.70 ± 0.58 | | |
| | Old bus | 3.43 ± 1.7 | 3.71 ± 2.3 | 47.14 ± 28.8 | | |
| | LDDV | 3.35 ± 2.20 | 2.03 ± 1.13 | 35.82 ± 21.4 | | |
| | HDDV | 2.20 ± 1.05 | 2.50 ± 1.43 | 31.0 ± 15.80 | | |
| | | 0.920 | nd | nd | Ban-weiss et al., 2008 | Roadside measurement |
| | | 0.5116 | nd | nd | Liggio et al., 2012 | Chasing measurement |



**Table 4: Average Particles EFs for Diesel and Gasoline of Road Traffic**

| Emission source | EF (g/kg-fuel) | | | OC/BC (%) | Reference |
|---|---|---|---|---|---|
| | BC | OC | TPM | | |
| GASOLINE ROAD | 0.62 ± 0.49 | 1.10 ± 0.77 | 7.0±2.8 | 177 | This work |
| | 0.15 | 0.73 | nd | 486 | Liousse et al., 2014 |
| DIESEL ROAD (mean) | 3.1 ± 1.9 | 2.14 ± 1.2 | 34.70 ± 20 | 69 | This work |
| | 5 | 2.5 | nd | 50 | Liousse et al., 2014 |
| Two Wheels ROAD | 2.13 ± 0.42 | 28.46 ± 0.4 | 420.52 | 1336 | This work |
| | 2.31 | 30.56 | nd | 1323 | Assamoi and Liousse, 2010 |



**Table 5: BC and OC EFs for Two Wheeled vehicles of our study and those of the literature**

| Emission source | Type | EFs (g/kg-fuel) | | | OC/BC | Reference |
|---|---|---|---|---|---|---|
| | | BC | OC | TPM | | |
| Two-Wheeled two-strokes vehicles | Recent | 2.26 ± 1.4 | 25.71 ± 1.1 | 238.3 ± 193 | 11.37 | This work |
| | Old | 3.45 | 124.21 | 883 | 36 | |
| | Mean | 2.74 | 65.11 | 496 | 23.8 | |
| Two-Wheeled four-strokes vehicles | Recent | 0.11 ± 0.01 | 0.45 ± 0.13 | 5.37 ± 4.64 | 4.1 | |
| | Old | 3.66 | 25.46 | 500 | 7 | |
| | Mean | 1.53 | 10.46 | 203 | 6.8 | |
| European Two-Wheeled vehicles | | 0.71 | 16.25 | nd | 22.89 | Bond et al., 2004 |
| European Two-Wheeled two-strokes vehicles | | 0.28 | 7.36 | nd | 26.28 | Guillaume and Liousse, 2009 |



**Table 6: EF for open solid waste burning of this study and those literatures**

| Emission source | EF(g/kg dm) | | | Reference |
|---|---|---|---|---|
| | BC | OC | TPM | |
| Waste burning | 2.8 ± 3.3 | 6.4 ± 4.6 | 87.9 ± 32.9 | This work |
| | 0.7 | 5.3 | nd | Christian et al., 2010 |



**Table 7: EF values of VOCs species for the studied sources : Heavy Duty Diesel Vehicles (HDDV), Light Duty Diesel Vehicles (LDDV), Light Duty Gasoline Vehicles (LDGV), Two-wheeled 2-strokes vehicles (TW 2T), Two-wheeled 4-strokes vehicles (TW 4T), Fuel wood (FW), Charcoal (CH), Charcoal Making (CHM) and Waste Burning (WB)**

| EF (g/kgdm) | HDDV | LDDV | LDGV | TW 2T | TW 4T | FW | CH | CHM | WB |
|---|---|---|---|---|---|---|---|---|---|
| heptane | 0.23±0.16 | 0.18±0.13 | 0.44±0.15 | 473±371 | 13.7±6.10 | 0.04±0.04 | 0.55±0.40 | **2.93±2.70** | 9.34±11.6 |
| octane | **0.74±0.46** | 0.21±0.10 | 2.36±0.40 | 470±423 | 8.09±4.00 | **2.38±2.30** | 0.87±0.50 | 2.25±1.70 | 6.31±9.10 |
| iso-octane | 0.09±0.07 | 0.21±0.15 | 0.04±0.03 | 204±217 | 0.74±0.17 | 0.37±0.60 | 0.42±0.59 | 0.14±0.04 | 0.40±0.60 |
| benzene | **0.68±0.27** | 5.60±2.80 | 4.78±0.40 | 379±279 | 32.0±8.50 | 2.00±1.98 | **8.64±12.0** | **4.20±0.12** | **19.1±19.0** |
| toluene | 0.58±0.17 | 3.10±1.80 | 34.7±18.6 | 1134±830 | 95.0±32.2 | 1.53±1.52 | 3.60±4.30 | 4.64±2.00 | 35.5±45.6 |
| m+p-xylene | **0.70±0.26** | 1.06±0.27 | 63.4±1.07 | 1334±810 | 56.3±21.0 | 1.43±1.42 | 2.17±2.04 | 2.63±1.70 | 5.50±8.60 |
| o-xylene | 0.32±0.15 | 0.56±0.22 | **30.1±3.43** | 793±536 | 25.0±9.33 | 0.16±0.47 | 0.51±0.50 | 0.73±0.22 | 0.37±0.43 |
| ethylbenzene | 0.25±0.11 | 0.82±0.37 | 15.6±3.15 | **814±590** | **40.6±16.1** | 0.24±2.07 | 1.74±2.07 | 0.80±0.25 | **27.8±34.3** |
| 135-TMB | 0.33±0.21 | 0.52±0.14 | 10.8±0.56 | 484±386 | 9.17±4.00 | 0.07±0.02 | 0.02±0.02 | 0.35±0.14 | 2.01±2.88 |
| 124-TMB | **1.29±0.96** | **1.73±0.34** | **34.6±0.84** | **1122±729** | **30.5±13.9** | 0.06±0.08 | 0.11±0.08 | 0.94±0.32 | 1.75±2.12 |
| 123-TMB | 0.50±0.40 | 0.71±0.14 | 5.84±0.36 | 309±195 | 6.62±3.12 | 0.65±12.7 | **7.74±12.7** | 1.58±0.55 | 0.57±0.77 |
| isoprene | 0.02±0.02 | 0.06±0.06 | 0.41±0.35 | 28.3±28.6 | 1.97±0.84 | **1.69±0.20** | 0.20±0.20 | 0.70±0.25 | 2.67±4.32 |
| limonene | 0.07±0.06 | 0.02±0.01 | 0.00±0.00 | 7.03±10.0 | 0.08±0.07 | **1.91±1.87** | 0.20±0.20 | 0.30±0.28 | **68.3±77.3** |
| a-pinene | 0.04±0.02 | 0.00±0.00 | 0.01±0.01 | 28.7±29.3 | 0.13±0.13 | 0.00±1.50 | 0.83±1.40 | 0.04±0.01 | 0.21±0.37 |
| b-pinene | 0.06±0.05 | 0.01±0.00 | 0.05±0.03 | 17.4±22.0 | 0.15±0.01 | 0.03±1.72 | 0.99±1.72 | 0.05±0.04 | 1.02±1.20 |



**Table 8 : EF for fuelwood burning of this study for ground field and combustion chambers measurements**

| Measurements | Methods | EF (g/kg dm) | | | | BC/TC |
|---|---|---|---|---|---|---|
| | | BC | OC | OC/BC | TPM | |
| Ground Field | Filter | 1.216 | 13.11 | 10.78 | 55.22 | 0.085 |
| Lannemezan | Filter | 0.078 | 1.13 | 14.49 | 6.658 | 0.064 |
| Edinbugh | HF1 filter | 0.095 | 0.557 | 5.86 | 2.790 | 0.146 |
| | hF1 filter | 0.049 | 0.356 | 7.26 | 2.194 | 0.121 |
| | HF1: AMS, SP2 | 0.444 | 0.742 | 1.67 | nd | 0.374 |
| | hF1: AMS, SP2 | 0.352 | 0.674 | 1.91 | nd | 0.343 |




**Table 9: Relative contribution of EF (BC) and EF (OC) per size classes to total size**

| Size-class | EF (BC) | EF (OC) | BC/TC |
|------------|---------|---------|-------|
| PM10 | 86% | 72% | 0.064 |
| PM2.5 | 81% | 72% | 0.062 |
| PM1 | 82% | 59% | 0.078 |
| PM0.25 | 77% | 51% | 0.094 |