# Peer review of "Aerosol and VOC emission factor measurements for African anthropogenic sources"

_Atmospheric Chemistry and Physics, 2017_

## Referee Comment (RC1) · Anonymous Referee #1 · 5 Dec 2017

**General Comments**

In this manuscript Keita et al. describe a set of field measurements in West Africa to better characterize emissions from several major emission sources specific to this region. Total particulate matter was collected on quartz filters and volatile organic compounds (VOCs) were sampled using sorbent tubes from all emission sources studied. Emission factors for organic carbon (OC), elemental carbon (EC, called black carbon in the manuscript), total particulate mass (TPM), and speciated VOCs were determined for the following emission sources: several African vehicles of various ages, trash burning, combustion of two wood fuels, charcoal burning, and charcoal making. Combustion emissions from a subset of the fuels studied in the field campaigns were also measured in the lab to gain more detailed information on particle size distributions

of the particulate emissions.

Given the extreme dearth in emissions data available that is relevant to Africa, this work is important and should be published to improve global and African emission inventories. However, the manner in which this work was presented in this manuscript does not provide convincing enough rationale that the work has enough atmospheric relevance to merit publication in ACP specifically. Authors would first need to give a clearer picture of the state of the science regarding African relevant emissions measurements from major sources and then show how their work significantly improves our understanding of African pollutant emissions and their environmental impacts. As part of this, the EF literature comparisons with this work can focus more on previous emissions measurements that are more relevant to African sources instead of seemingly randomly selected emissions studies of sources that have little to no relevance to African emissions. One suggestion to better demonstrate the environmental impact of this work is to make specific recommendations to update the African Regional Inventory. Another suggestion is to use relevant activity data and emission factors from this work to calculate total particulate and speciated VOC emissions for West Africa and compare the environmental impacts of major West African emission sources. Another possible route I would suggest to increase the scientific impact of this paper to merit publication in ACP is to expand in greater detail the discussion of the more scientifically novel speciated VOC/IVOC measurements while giving a more concise discussion of the OC/EC measurements. The VOC measurements are not only scientifically impactful due to the lack of this kind of measurements for Africa, but also the measurement/analytical methods used in this study and the specific compound list including difficult to measure IVOCs and carbonyls are also of great scientific interest for atmospheric/emissions scientists. The authors focus much of the paper on OC/EC measurements (and a surprising amount to predictable MCE values), discussing these measurements in great detail for each emissions source, whereas the VOC measurements are essentially glossed over. In fact, only a small fraction of the emission factor data generated from the VOC measurements (15 out of 50+ VOCs) in this study was

actually presented (including SI). How can others improve emissions inventories and assess the atmospheric impacts of these emissions if the emissions data is not reported? This seems to run counter to the objectives of this work.

The final major concern that needs to be dealt with before being considered for publication is that the authors do not provide enough detail for the reader to understand how the measurements were conducted. The only exception involves the chamber measurements, which only make up a minor part of the results but are discussed in great detail in the methods. More specific suggestions to add detailed information on the emissions measurements are given in the specific comments below.

Specific Comments

Title. Aerosol measurements weren't presented. I suggest changing "Aerosol" to "Particle". It would also be helpful to make location more specific, i.e. West Africa. Abstract

Line 17. Acronym is not used again, so it's not needed. Please put acronyms in parantheses.

Line 20. THE "NM" in NMVOCs is redundant. VOCs is sufficient.

Line 23. Particles were collected not aerosols. State what is measured by these methods.

Line 24. What is meant by systematic? What kind of sampling, what analysis? What type of wood was used?

Line 25. Be clear that calculations were based on mass fuel burned not dry matter?

Line 28-29. A comparison of PM EFs and VOC EFs is not useful.

Line 33. This statement is too vague.

Introduction: This section needs a summary of previous emissions measurements for major emission sources in Africa/West Africa.

Line 11. This statement suggests that no significant pollution related health impacts are to be expected until 2030. Is this what is meant?

Line 13-14. According to Louisse et al. 2010: "Predominant emissions in the BC class are related to use of diesel fuels, animal waste, fuelwood, charcoal making and coal." "OC emissions are mostly impacted by animal waste, charcoal making, fuelwood and two-wheeled vehicle fuels." Please give a more detailed discussion of African emission sources.

Line 15. What type of carbon is referred to here- gas or particle carbon or both?

Line 16-18. The focus of this paper is on Africa, so why discuss global energy use?

Line 19. Second hand is not an indication of age.

Line 20. What is the reference - Robert 2007a or b? Peltier is not in the reference list.

Line 22. This statement is awkward. Why is this important and how big of a problem is this?

Line 25. Multiple sources are mentioned, but only discuss one. Please make clear that this is discussing trash burning not animal waste burning?

Line 32. This statement is vague. Can you be more specific? What is meant by not well documented EFs? Activities?

Line 1. Vague sentence – please clarify.

Line 6-9. Aren't all published EFs literature EFs? What does this mean? What does very rare mean? What about nontraffic sources? What studies have been done on EF measurements relevant to Africa?

[Figure]

Line 9. Please add a reference for Africa Regional Inventory.

Line 11-16. When EF measurements are so rare, differences between methods is not particularly important.

Line 16. This suggests that particulate OC/EC is the only pollutant worth measuring in emissions. Is this what is meant?

Line 18-20. Sentence is awkward. "is" should be "are" See above comment on NMVOC. Particulate matter should lower case.

Line 22. Acronym should be in parentheses. Work Packages should be capitalized.

Line 26. Primary is redundant.

Line 31-32. Section 1 and 2 don't correspond to sections in the paper. Please state whether this work is being presented in other papers and which measurements this paper will cover and which will be covered in other papers.

Methods. It would be more helpful if measurements were discussed before calculations.

Line 7. Should it be Ward et al. 1993?

Line 15. This equation is incorrect to me as it is currently written. I do not understand why the MCE was added to this equation; it adds unnecessary terms that make the equation no longer correct. If this is how the emission factors were calculated, I believe all the EF values in this manuscript need to be corrected by removing the MCE term. If the authors feel this is correct, they need to provide an explanation and a reference where this equation was used for EF calculations.

Line 3. List countries.

Line 5. Show site on a map.

Line 5,8. What is meant by "measurement points" and "samplings" - eight locations or eight tests from different plumes, or eight samples from the same plume?

Line 10-11. Please elaborate on "different characteristics". Does "services" mean electricity?

Line 12-14. More details are needed for these measurements. How much fuel was used? Was the fuel prepared, e.g. dried before use? Were any fuel properties tested such as moisture content? What were the cookstove/kiln characteristics, designs and models; show pictures/diagrams of these. Show sampling setups. Show location where these measurements were conducted on a map if they were field measurements. How exactly were the burn experiments initiated and conducted- what were the exact protocol? Please give this information for each emissions sources studied.

Line 20-23. State make/models, model years, mileage, engine specifications of all vehicles tested. Please list the tests conducted and number of replicate tests for each vehicle. What fuel was used and where was it sourced? How exactly were the vehicles operated during testing? Was a dynamometer used? What was the specific driving protocol? Show a diagram of the sampling setup.

Line 26. State Equation 1 or Eq. 1 instead of "equation".

Line 30. How often was the instrument's calibration checked? Was this done in the field?

Page 6.

Section 2.4. Misleading title. The combustion didn't occur in the chamber, the chambers were just used for dilution. Why were these chamber experiments conducted? Were they part of this initiative? Please include rationale in the Introduction.

Line 1. Why was air used? I believe heated air (oxygen) damages the sorbent.

Line 4. State the pump vendor/model.

Line 5. Why was no size cutoff selected? State quartz filter vendor PN, size

Line 9. Which sources?

Line 14. Does this mean room temperature or lower? What is meant by "homogeneous"? There are gas and particles, thus it is not homogeneous. Do you mean well mixed? Which stove was used for the combustion – give more details or reference with details.

Line 24. What were the conditions in this study?

Line 28. This statement is misleading. It has not been made clear that these chamber measurements have already been discussed in detail in Haslett et al. Why are these measurements being discussed again in this paper? If the sampling details are given in this reference, they do not need to be repeated in detail here.

Line 3. This is not a complete sentence.

Line 10-14. This section describes the sampling media that should be discussed in Section 2.3.

Line 11. Please change "absorbent" to "sorbent".

Line 12. Please elaborate in more detail why these two types of tubes were used. Did these tubes capture different volatility ranges or different classes of VOCs? State vendor and part number for tubes or if hand packed, state so.

Line 14. Where was LAMP located? Need to discuss how samples were transported and hold times between sampling and analysis for all samples. Has it been confirmed that stability times for all target compounds on the tubes exceed the hold times in this study? Were any quality control samples, such as field blanks or controls, included?

Line 21,27. Please list the compounds that were measured and discuss why these compounds were selected. If they are listed in SI, note here. Were there overlapping VOCs? If so how did they compare between the two tube methods? Also provide method detection limits for each VOC and each instrument in SI.

Line 25. Why were different types of tubes analyzed in different labs? Spell out acronym.

Line 26,27. State exact number of compounds. See above comment.

Line 28. This is misleading. For this study, only TPM is measured.

Line 31,32. This should be called elemental carbon instead of black carbon. Correct the manuscript accordingly. Also, these are particles and not aerosol measurements – please correct sentence.

Line 2. By "the thermo-optical method", this refers to the IMPROVE method? Isn't the method used in this study also a thermo-optical method? What was the rationale behind using this less well validated method as opposed to IMPROVE or NIOSH? Which samples were used in the comparison; were they from this study or other samples? Was the analysis done on the same instrument? What instrument was used, and how was it calibrated?

Line 4. What is considered suitable?

Line 15-18. "Residential sources" is too vague. Need complete sentences between semicolons.

Line 20. Please change "aerosol" to "particulate" or "particle". Tables should be numbered in order of when they are first mentioned in the text. This Table should be Table 1. Only 15 VOCs are listed. Why is only a subset of compounds listed here? It is mentioned that three tests per source were performed – how many samples per test

were taken? Please list exactly how many tests (and samples taken per test) were conducted for each source. This could go in the SI.

Line 21-22. Why geometric mean? Which sectors are being referred to?

Line 26. Fossil should not be capitalized.

Line 28. Please state the biofuels.

Section 4.1.1. I suggest removing Section 4.1.1 as much of it is quite predictable and not scientifically interesting, and instead discuss MCEs along with the EFs in other sections.

Line 1. Why are MCEs summarized in a Table and Figure – this is redundant. The MCEs can easily be summarized in Tables 2 and 3 with the emission factors. Modified combustion efficiency should be lower case. Both MCE and CO/CO2 are both combustion indicators – it is redundant to discuss both.

Line 2-4. The meaning of this sentence is not clear. Please clarify.

Line 14. BC/TC ratios are mentioned here before showing any results on their emission factors. This relationship with MCE should be brought up during discussion of "BC" emission factors.

Line 19. Title is too vague. Residential heating? Residential cooking? Is charcoal making a residential source? For these subsections (4.1.2-4.1.4) only particulate EFs are discussed, so make that clear in the subsection titles.

Line 20,22. EF should be plural – EFs. Please check the entire manuscript for this error. Please reference Equation 1 consistently.

Line 21. Table should be capitalized and it should be Table 2. Please check the rest of the manuscript for this error.

Line 22. "Dry matter" is misleading. The calculation is per kg fuel burned not dry matter. Please make this clear.

Line 23-25. Is there any evidence behind these assumptions?

Line 26-28. This is a contradictory statement. Are they in agreement or higher than literature? What fuels did these other studies test? Have emissions from Hevea, Iroko or any other African-specific fuel been measured previously?

Page 10.

Line 1. This is too vague. What studies?

Line 2-4. I do not agree with this argument. If carbon is left in the fuel, it is not burned and not emitted - so it is not included in "mass of fuel burned" using the mass balance method. Therefore, I do not believe adding this additional multiplication factor is correct.

Line 11. This should be Table 2. Please check entire manuscript for this error in Table numbering.

Line 13. This statement is contradictory, as it was earlier stated that CHM conditions in Cachier et al. were similar to this study.

Line 18. Recent and old are not age groups and are thus meaningless adjectives. Please state model year groups.

Line 19-22. Which "literature values" are being discussed in Line 19? Why were these particular studies selected for comparison given the large number of vehicle emission studies available? Do they have any applicability to West African vehicle fleet? If not, then what is the reason for the literature comparison? State units for EF values in Line 19.

Line 23. Please explain the meaning of "coherent with Fig 4". Figure 4 does not show this statement to be true for diesel EFs in this study, if I understood what was mean.

Line 27-31. This is probably one difference of lesser importance than vehicle age and lack of emission controls. Therefore, it is unclear why it is emphasized over other more important differences. Provide reference for evidence supporting these statements/assumptions in Lines 28-31. What is meant by "park vehicle"?

Lines 32-33. Include units.

Line 3. Why compare to HDDVs? They are not relevant.

Line 4. Please report all EFs for individual vehicles in the SI. I don't see how these average values are helpful and why they require a separate table. How are these values expected to be used?

Line 8. Why are these values compared to European inventory? What is the usefulness if this comparison?

Line 11. This statement is misleading. Approximate agreement of OC/EC ratios does not justify this general statement.

Line 13-15. This is the most relevant literature comparison I've seen in this section, so this needs to be discussed in much greater detail.

Line 16. Weren't all the vehicles classified the same way?

Line 27-29. See comment for line 13-15.

Line 2. Instead of "bad" I suggest "less efficient".

Line 11-13. Were these different phases observed during the same test/sample or different tests? Can you include a Figure showing real time $CO_2$, CO measurements during a burn to demonstrate this? Please expand in greater detail.

Line 23. Why are only these VOCs shown? Did they have the highest EFs for all

sources? Where are the emission factors for the other VOCs?

Line 25-27. Aren't all VOCs important for atmospheric reactivity to some degree?

Line 29. Given that no background samples are taken, how is it known that these are not solely from biogenic emissions? Please discuss.

Line 31-32. These analytical uncertainties between methods needs to be discussed in greater detail in the Methods section. Please clarify the statement "different sources associated to the emission sector analysed".

Line 1. What is meant by "most important"?

Line 10. What do these values represent?

Line 20. Since a full VOC target list has not been given, it is impossible to evaluate whether enough compounds were measured to be representative of the different species groups.

Line 26. If the aldehydes were a large contribution, why were they omitted from Table 7 or discussed earlier in this section?

Line 29. The relevance of these particular studies for comparison needs to be explained. They do not seem to be at all applicable to this work, so this comparison is not helpful.

Line 1. I am not convinced that this is true based on the discussion above.

Lines 5-10. This should be moved to the Methods section on EF calculations.

Line 14. There are many possible reasons for these discrepancies. Were the fuels burned using the same cookstoves under the same protocol? Were the MCEs similar? Were the fuels of the same moisture and carbon content? Perhaps the sampling meth-

ods/particle losses were different? Dilution can certainly affect OC (and this should be discussed further), but how will it affect EC EFs? Please elaborate.

Section 4.2.1. These measurements have already been compared in the previous paragraph. Why is there subsection here in the middle of this continuing discussion on the topic?

Line 26. If CO/CO2 ratios are that different that is a measure of combustion efficiency, then aren't the combustion conditions also very different between field and lab? Why are BC/TC ratios used as a combustion efficiency indicator instead of MCE values?

Line 31. Filter method in this study measured EC not BC. There could be differences due to the different methods that should be discussed here.

Page 15.

Section 4.2.2. What can be said about size distributions from the other emission sources in this study? Please discuss this issue for each emissions source and give appropriate references.

Line 7. Diesel engines are not relevant.

Line 17. This sentence is too vague.

Line 20. Please state that it is particle mass.

Line 21. What EFs are being referred to? None of the EFs were shown as a function of vehicle age or maintenance, so this conclusion has not been confirmed by the previous discussion.

Line 24. "Published EF values" for what specific source?

Line 26. EFs of all VOCs have not been provided, so there is no evidence given that this statement is correct.

Line 30-32. The meaning of this sentence is unclear.

Page 16.

Line 1. State the fuel used for this measurement.

Table 9. Please state the emission source/fuel burned in the caption.

[Figure]

---

## Referee Comment (RC2) · Anonymous Referee #2 · 28 Feb 2018

This work presents the Emission Factors (EFs) from fuel combustion in West Africa estimated from field and combustion chamber measurements. EFs were estimated for black carbon (BC), primary organic carbon (OC), total particulate matter (TPM), and 50 non-methane volatile organic carbon (NMVOC) species. In addition, measurements in combustion chambers were used to estimate particulate EFs by size, namely for PM10, PM2.5, PM1 and PM0.25. This work was conducted within the framework of the DAC-CIWA (Dynamics-Aerosol-Chemistry-Cloud Interaction in West Africa) FP7 program. Field measurements were conducted in two places of West Africa, namely Abidjan in Ivory Coast and Cotonou in Benin, while measurements in combustion chambers were conducted in Toulouse, France, and in Edinburgh, United Kingdom. The considered emission sources are wood and charcoal burning, charcoal making, open waste burn-

ing and vehicles including cars, trucks, buses and two-wheeled vehicles. As mentioned by the authors in the introduction, one of the sources of uncertainties in emission inventories are uncertainties in the emission factors and therefore research contributing to reduce these uncertainties are necessary to better quantify the impact of given source sectors on air quality and climate. However, although I recommend this paper to be published in ACP some important aspects need to be addressed before its publication.

General comments

1. The authors mention the role of uncertainties in the uncertainties of emission inventories (EI) in the introduction, however they do not address this issue again in the rest of the manuscript. How does this work contribute to reduce uncertainties in current EI in Africa? What are the main sources of uncertainties in EF in the region?

2. After reading the article it is not clear to me what the real contribution of this article is. The authors make an attempt of this in the last paragraph of the conclusions but not supported in the text. The authors should make an effort and discuss the implications of their results for current Emission Inventories. Are emissions currently over or underestimated if these new EFs are considered?

3. It is not clear to me that the title is the most appropriate considering the scope of the study. Measurements are conducted only in two places of tropical West Africa but the authors claim (not explicitly) that the results are valid for Africa. How representative of other countries in Africa and/or West Africa are these results? Can they be extrapolated for the entire region? The authors show with their results that EF are sensitive to multiple factors therefore applying these emission factors to other countries is not straightforward. The scope of the study should be consistent with the title and the content of what is presented. It is not only a matter of the title but also how the data are presented and their representativity.

4. The authors are not thorough when presenting the results and formulation is unclear in certain places. Authors should review the manuscript for consistency and improve

formulation. For instance, in page 9 the numbers provided from lines 5 to 7 do not match the numbers given in the table indicated in the text. Also in page 11, line 18, the values in the text are approximated to one decimal while in the table it is with two. Authors should review the text and correct inconsistencies between the data provided in the text and the ones given in the tables and figures.

5. The authors make an attempt to put the results in perspective in the last paragraph of the conclusions but without elaborating on it in the text. The authors should elaborate on the impact of the EFs and provide more evidence on the importance of their findings for current and future emission inventories.

6. Authors should make sure that all methodological description is presented and described in section 2. At present, section 4 also includes important methodological aspects in terms of composition of sources and how the data are combined to obtain total EFs per source. This should be moved to section 2 and more information should be presented. It should be made clear to the reader how the data are aggregated.

Specific comments

Page 3, lines 18-21: formulation is unclear and should be improved.

Page 3, lines 22-25: Unnecessary information is provided on worck packages of the DACCIWA project but no general information of the project itself is provided. I suggest removing the unnecessary information and include a short description of the project and then linking it to the work presented, without necessarily tying it to a WP.

Page 3, lines 31-33: Section numbers are not correct, in addition the paper has five sections and only two sections are presented and described. The authors should complete this paragraph and make sure that section numbering is consistent.

Page 4, line 1: remove "in the frame of the DACCIWA WP2" since this information has already been provided and information on the WP is not really informative and relevant.

Page 5, lines 4-7: formulation is unclear and should be improved.

[Figure]

Page 5, line 31: What is the unit of the provided uncertainty, is it % or ppm?

Page 5, lines 2-23: The description of the field measurements needs to be improved. It is unclear weather the same measurements were conducted in each on of the 3 field campaigns or if they were different. For instance, when it is said that eight measurements were carried out at Akouedo was it in each one of the three campaigns or only in some of them. Please clarify, manuscript should be clear in terms of the measurements that were conducted in each campaign.

Page 5, line 27: Provide a reference of the instrument QTRAK-7575.

Page 6, section 2.4: Section 2.2 indicates that two types of African hardwood were tested in combustion chambers, namely Hevea and Iroko, however in section 2.4 only Hevea is mentioned and it is not clear whether Iroko was tested at all. Was Iroko included in the study? Results corresponding to wood burning correspond only to Hevea? Please clarify.

Page 7, line 9: Section 3 should in fact be part of section 2. Why do the authors consider it should go in a section by itself?

Page 7, line 26: . . .in Detourny et al. (2011) and Ait-Helal et al. (2014).

Page 8, line 20: First table mentioned in the text is Table 7! But no mention was done before to the previous 6 tables. Numbering of the tables should be done in order they are referenced in the text.

Page 8, lines 20-21: What exactly is meant by this? Does it mean that in addition to the number of measurements indicated in section three additional measurements were conducted to reflect reproductibility or are these three measurement part of the total number of measurements conducted? This should be made clear and should be included in section 2 and not in the results and discussion section.

Page 8, lines 21-22: What are the arithmetic and geometric method the authors mentioned, is it just averaging? Again, this should be made clear and included in section 2

where the measurements are described. See general comment made above.

Page 8, line 26: DL and MO have not been defined in the text. They are defined in the caption of figure 3 but should also be defined in the text.

Page 8, lines 28-29: How was the MCE of biofuels from 0.6 to 0.9 from Iroko wood obtained? Was it from the combustion chambers or from the literature? This should be clarified.

Page 9, lines 1-2: the $\Delta CO/\Delta CO2$ ratio has not been introduced as a quality indicator before, the statement should be reformulated.

Page 9, lines 2-4: remove the coma after 0.32 and replace "showing" with "show".

Page 9, lines 5-7: Values provided in the text do not correspond to values in Table 1!!! This needs to be corrected and the authors should make sure that the numbers given in the text are consistent with those in the tables and figures.

Page 9, line 15: Three sources are mentioned but only two symbols are provided in parenthesis. Furthermore, the authors make a too simple analysis of the results presented in Figure 4. Above 0.2 BC/TC one could agree with the authors that the larger the $\Delta CO/\Delta CO2$ ratios the smaller the BC/TC ratio. However, below 0.2 BC/TC the data suggest that regardless of the $\Delta CO/\Delta CO2$ ratio, the BC/TC ratio is mostly constant. The authors should elaborate on these two regimes and explain the reasons behind it.

Page 9, line 24: What calculations do the authors refer? How were these calculations made and by whom? Authors should provide more information about this.

Page 9, line 27: Hasn't there been anything more recent than the references provided in this line? The authors should look for more recent EFs estimates.

Page 9, lines 27-28: The literature review of EF in West Africa is not thorough enough to make this kind of statement. Make the statement relative to the studies included in

table 2 and not general.

Page 9, lines 31-33 and Page 10, line 1: Very little is said about Charcoal EF in contrast to wood and charcoal making. The authors should elaborate more on Charcoal. Also, the authors should make it a separate paragraph from wood and not split it on two paragraphs as it is now.

Page 10, lines 4-5: Put parenthesis after years of publication for each reference.

Page 10, line 19: Again the values given in the text do not correspond to values provided in Table 3 and here again it's a matter of significant numbers considered in the text and in the table. The Authors should check the paper for consistency and use the same criteria with regards to significant numbers when presenting results.

Page 10, line 19: I do not agree that the estimated EF(BC) for new LDGVis is within the range of literature values. It corresponds in fact to the lower limit of the range of values provided in the table. The authors should make the statement consistent with the data in the table. Also, it is unclear whether the literature values presented in the manuscript are from the same country or region or from elsewhere in the world. Although the measurement method is presented, not the country where the study is conducted and this information should be provided or at least considered when comparing the estimates.

Page 10, line 30-31: Where are these percentages of old and recent vehicles taken from? Are they based on statistics from the literature, governmental documents, etc? The authors should explain where these numbers come from. Furthermore, this should be included in section 2.

Page 11, line 7: Again where are these numbers of 77% of light duty vehicles and 23% heavy duty vehicles? Same as comment before, these numbers need to be justified somehow and also moved to section 2.

Page 11, lines 8-15: Again values presented in the text do not match those in the corresponding table referenced in the text (Table 4). Furthermore the data are compared

to different COPERT EURO Standards without even presenting them and explaining them. Furthermore, why are these data included in table 4? I would strongly suggest the authors to rewrite this analysis including the different COPERT values in table 4.

Page 11, line 18: Again value of 26.0 +- 1.10 g/kg given in the text is not the same as the one given in table 5 (25.71 +- 1.1). If the authors decide to use a certain criteria of significant numbers for the table, they should use the same for the data presented in the text. There is absolutely no reason why numbers should not be exactly the same. Again, please be consistent!

Page 11, lines 19-20: What exactly is meant in the sentence "The same difference is observed for old...". The difference in EFs(OC) in fact for old cars between two and four stroke is much smaller than for recent ones. The authors should correct the analysis and make it consistent with the data presented by them.

Page 11, lines 20-21: The statement that two stroke engines emit more OC than four stroke engines is made only for old vehicles, why is not the same analysis done for new vehicles? Also, the data to support that claim is not the OC/BC ratio but the actual EF, authors should correct this.

Page 11, lines 27-29: Where are the percentages of two and four stroke engines taken from? As before, these numbers need to be justified and also be moved to section 2. Also, the Table 5 is referenced here, but shouldn't it be table 4? Finally, rather than saying that the values of this work are in agreement I would suggest to reformulate and say they are comparable.

Page 15, lines 21-22: Authors conclude on the dependence of EF from traffic to vehicle age and maintenance, although the latter was not included in any of the analysis presented. The authors should base the conclusions to the results presented in the article. I suggest maintenance is removed from the conclusions or the corresponding data to support that claim are provided.

Page 16, line 6: Authors claim that EFs obtained are representative of African sources but do not provide any evidence or analysis of this in the paper nor a reference to support this claim. I suggest the authors provide some evidence to support this statement or remove it from the conclusions.

Page 16, lines 7-9: "This unique database..." why is the database unique? No mention to its uniqueness has been made before. How will it improve emission inventories in Africa? What will be the impact of these database on emission estimates? Again, no mention of the impact of these EFs on emission inventories is made in the text to support this. Finaly, how will this new EFs help decision makers? Wouldn't this usually be come from new emission inventories? These kind of statements, although tempting should be supported by the facts presented in the text which I believe are not. Please reformulate.

---

## Author Comment (AC2) · 23 Apr 2018

Dear Editor, First, we would like to thank the reviewers for their positive comments and their contribution to improve the quality of this document. All the questions were treated and our document was fully revised taking into account all the reviewers comments. The paper structure was modified as a result of the different suggestions and remarks, resulting in different figures, tables and section numbers. Please find attached a point-by-point response to the reviewers' questions.

Please also note the supplement to this comment:
https://www.atmos-chem-phys-discuss.net/acp-2017-944/acp-2017-944-AC2-supplement.zip

---

## Author Response (AR1)

**ACP-944-2017**

**Old title "Aerosol and VOC emission factor measurements for African anthropogenic sources"**

**New title "Particle and VOC emission factor measurements for anthropogenic sources in West Africa"**

By Keita et al.

**Response to Reviewer's comments**

Dear Editor,

First, we would like to thank the reviewers for their positive comments and their contribution to improve the quality of this document. All the questions were treated and our document was fully revised taking into account all the reviewers comments. The paper structure was modified as a result of the different suggestions and remarks, resulting in different figures, tables and section numbers. Please find attached a point-by-point response to the reviewers' questions.

Referees' comments (**in black**), author's response (**in blue**) and changes in the revised manuscript (**in red**).

**Reviewer's comments**

**Referee #1**

**Anonymous Referee #1**

General Comments

In this manuscript Keita et al. describe a set of field measurements in West Africa to better characterize emissions from several major emission sources specific to this region. Total particulate matter was collected on quartz filters and volatile organic compounds (VOCs) were sampled using sorbent tubes from all emission sources studied. Emission factors for organic carbon (OC), elemental carbon (EC, called black carbon in the manuscript), total particulate mass (TPM), and speciated VOCs were determined for the following emission sources: several African vehicles of various ages, trash burning, combustion of two wood fuels, charcoal burning, and charcoal making. Combustion emissions from a subset of the fuels studied in the field campaigns were also measured in the lab to gain more detailed information on particle size distributions of the particulate emissions.

Given the extreme dearth in emissions data available that is relevant to Africa, this work is important and should be published to improve global and African emission inventories. However, the manner in which this work was presented in this manuscript does not provide convincing enough rationale that the work has enough atmospheric relevance to merit publication in ACP specifically. Authors would first need to give a clearer picture of the state of the science regarding African relevant emissions measurements from major sources and then show how their work significantly improves our understanding of African pollutant emissions and their environmental impacts. As part of this, the EF literature comparisons with this work can focus more on previous emissions measurements that are more

relevant to African sources instead of seemingly randomly selected emissions studies of sources that have little to no relevance to African emissions. One suggestion to better demonstrate the environmental impact of this work is to make specific recommendations to update the African Regional Inventory. Another suggestion is to use relevant activity data and emission factors from this work to calculate total particulate and speciated VOC emissions for West Africa and compare the environmental impacts of major West African emission sources. Another possible route I would suggest to increase the scientific impact of this paper to merit publication in ACP is to expand in greater detail the discussion of the more scientifically novel speciated VOC/IVOC measurements while giving a more concise discussion of the OC/EC measurements. The VOC measurements are not only scientifically impactful due to the lack of this kind of measurements for Africa, but also the measurement/analytical methods used in this study and the specific compound list including difficult to measure IVOCs and carbonyls are also of great scientific interest for atmospheric/emissions scientists. The authors focus much of the paper on OC/EC measurements (and a surprising amount to predictable MCE values), discussing these measurements in great detail for each emissions source, whereas the VOC measurements are essentially glossed over. In fact, only a small fraction of the emission factor data generated from the VOC measurements (15 out of 50+ VOCs) in this study was actually presented (including SI). How can others improve emissions inventories and assess the atmospheric impacts of these emissions if the emissions data is not reported? This seems to run counter to the objectives of this work.

The final major concern that needs to be dealt with before being considered for publication is that the authors do not provide enough detail for the reader to understand how the measurements were conducted. The only exception involves the chamber measurements, which only make up a minor part of the results but are discussed in great detail in the methods. More specific suggestions to add detailed information on the emissions measurements are given in the specific comments below.

We thank Referee #1 for providing very useful comments and suggestions on the manuscript which, given the extreme scarcity of emissions data available and relevant to Africa, will improve emissions inventories over Africa. Following the referees' comments, many changes and reorganizations were made in the manuscript. A clearer picture of the state of the science regarding African relevant emissions measurements from major sources was given in the introduction. As suggested by the reviewer, the comparison of the EFs of this work with those of the literature now focused on the EFs measured in Africa in one hand and on the other hand on the EFs used in the emission inventories for Africa. Second, due to the lack of existing speciated VOCs EFs in Africa, we used IEA activity data and speciated VOCs EFs from this study to calculate emissions that were compared to those from EDGAR for Côte d'Ivoire. Finally, details were provided for the reader to understand how the measurements were conducted. The responses to the Referees comments on Specific Comments are found below.

Specific Comments

**Page 1**

Title. Aerosol measurements weren't presented. I suggest changing "Aerosol" to "Particle". It would also be helpful to make location more specific, i.e. West Africa. Abstract

Thank the manuscript title has been changed as suggested
Particle and VOC emission factor measurements for anthropogenic sources in West Africa
Line 17. Acronym is not used again, so it's not needed. Please put acronyms in parantheses.

This was done. Thanks

Line 20. THE "NM" in NMVOCs is redundant. VOCs is sufficient.

NMVOCs was replaced by VOCs

Line 23. Particles were collected not aerosols. State what is measured by these methods. This was corrected and the sentence has been edited as following

Particle samples were collected on quartz filters and analysed using gravimetric method for total particulate matter and thermal method for elemental and organic carbon.

Line 24. What is meant by systematic? What kind of sampling, what analysis? What type of wood was used?

Thank you. The word "systematic" has been removed, off-line sampling on sorbent tube was used and this sentence has been edited. We have used two tropical hardwood species: Hevea and Iroko

Line 25. Be clear that calculations were based on mass fuel burned not dry matter?

Thanks. The calculation was based on fuel burned, dry matter has been replaced by fuel burned

Line 28-29. A comparison of PM EFs and VOC EFs is not useful.

Thanks. This comparison was removed in the revised version of the manuscript

Line 33. This statement is too vague.

Thank you. This part was edited as following

The particle and VOC emission factors obtained in this study are generally higher than those in the literature whose values were discussed in this manuscript.

**Page 2**

Introduction: This section needs a summary of previous emissions measurements for major emission sources in Africa/West Africa.

The introduction has been considerably modified following the reviewers' comments. Changes are in red in the new version of the document.

Line 11. This statement suggests that no significant pollution related health impacts are to be expected until 2030. Is this what is meant?

This sentence was edited

Liousse et al., (2014) have shown that if nothing is done as quickly as possible, the climate and health impacts could significantly increase by 2030, when African pollution level could become higher than those in Asia.

Line 13-14. According to Liousse et al. (2010): "Predominant emissions in the BC class are related to use of diesel fuels, animal waste, fuelwood, charcoal making and coal" "OC emissions are mostly impacted by animal waste, charcoal making, fuelwood and two-wheeled vehicle fuels"

Please give a more detailed discussion of African emission sources.

Done. P3-line 20-24 in the revised manuscript introduction

In Africa, the most recent regional African inventory has shown that black or elemental carbon (BC or EC) emissions are dominated by the use of diesel fuels, animal waste, fuelwood, charcoal making and coal (Liousse et al., 2014). While, animal waste, charcoal making, fuelwood and two-wheeled vehicle fuels mostly affect organic carbon (OC) emissions. The authors also showed that West Africa has maximum emissions in the domestic and traffic sectors for EC, OC, carbon monoxide, nitrogen oxides and volatile organic compound species due to combustion of the fuels above mentioned.

Line 15. What type of carbon is referred to here- gas or particle carbon or both?

Here we referred to particle carbon and the sentence has been edited:

Andreae and Merlet, (2001) have already shown that domestic fires used for cooking are an important source of primary pollutants (gas and particle) worldwide, particularly in Africa.

Line 16-18. The focus of this paper is on Africa, so why discuss global energy use?

This paragraph was rewritten:

The main source of energy in households in developing countries are solid fuels such as charcoal, agricultural residues and wood (Wang et al., 2013) : in Sub-Saharan Africa, these biofuels represent approximatively 80% of the total energy consumption (Ozturk and Bilgili, 2015).

Line 19. Second hand is not an indication of age.

This was edited. Thank you

In addition, the traffic fleet in Africa is characterized by an aging fleet (more than 80 % are second-hand vehicles with 73 % older than 10 years old) (Kablan, 2010; Essoh, 2013).

Line 20. What is the reference - Robert 2007a or b? Peltier is not in the reference list.

Thank you, reference is Robert 2007a and Peltier et al., 2011 was added in the reference list.

Line 22. This statement is awkward. Why is this important and how big of a problem is this?

This was completed

In some African countries, it is also important to note the importance of two-wheeled vehicles (two-stroke or four-stroke engines) using a mixture of oil and gasoline derived from smuggling that is very polluting (Assamoi and Liousse, 2010).

Line 25. Multiple sources are mentioned, but only discuss one. Please make clear that this is discussing trash burning not animal waste burning?

Here it is the trash burning that is discussed, this line was rewritten:

In addition, very high levels of pollutants are associated to trash burning emissions and this source has not been well studied in Africa.

Line 32. This statement is vague. Can you be more specific? What is meant by not well documented EFs? Activities?

"However, none of the above-mentioned sources are well documented."

This sentence was edited as follow

However, EFs of carbonaceous particles and VOCs for the above-mentioned sources are not well documented in Africa, leading to a scarcity of emission factors from these sources.

**Page 3**

Line 1. Vague sentence – please clarify.

"The existing emission inventories are often global and involve many uncertainties, particularly in Africa (Assamoi, 2010)."

This sentence was edited :

Existing emission inventories for Africa extracted from global emission products use inadequate emission factors, which are not measured in Africa and consequently not relevant to specific fuels and combustions, and activity consumption data given by international agencies (e.g. UN, IEA). These inventories therefore include many uncertainties in Africa related to the use of such global data (Assamoi and Liousse, 2010).

Line 6-9. Aren't all published EFs literature EFs? What does this mean? What does very rare mean? What about non traffic sources? What studies have been done on EF measurements relevant to Africa? Thanks for the comments. Effectively our purpose was not clear. For us, the term 'literature data' meant that these EFs were issued from measurements occurring elsewhere than in Africa. This was clarified in the text. Moreover, "Very rare" has been replaced by "scarce".

This part was rewrite focus on EF measurement studies in Africa as suggested by reviewer.

Biomass burning EFs for a number of gaseous and particulate species have been compiled by Andreae and Merlet (2001) and Akagi et al. (2011), showing that many studies have been carried out in Africa since 1994. However, literature shows only a few EFs measurement studies on anthropogenic sources. Most existing works in that area mainly focused on biofuel combustion. For example, particles and gases EFs measurements for wood cooking fire have been performed by Brocard et al., (1996), Brocard and Lacaux, (1998) in Côte d'Ivoire and by Bertschi et al., (2003a) in Zambia. Particles and gases EFs measurements have also been performed by Bertschi et al., (2003a), Brocard et al., (1998), Kituyi et al., (2001) for charcoal cooking fire and by Pennise et al., (2001), Lacaux et al., (1994), Brocard and Lacaux, (1998) for charcoal making fire. Unfortunately, such studies are only done for a few pollutants. To our knowledge, EF measurements for traffic vehicles in Africa are very scarce. This means that even in the existing African Regional Inventory of Liousse et al., (2014), literature EFs issued from US or European emissions and not relevant for Africa have been sometimes used. Applying such values to Africa is a large source of uncertainties.

Line 9. Please add a reference for Africa Regional Inventory.

The reference for Africa Regional Inventory has been added.

Line 11-16. When EF measurements are so rare, differences between methods is not particularly important.

This part was removed. Thank you

Line 16. This suggests that particulate OC/EC is the only pollutant worth measuring in emissions. Is this what is meant?

Carbonaceous aerosol is the main compound of combustion aerosol. In spite of this importance, and in spite on the impacts of such particles, carbonaceous aerosol is not enough well documented. This is why we focused on these particles. Of course, it will be important in the future to also focus on the other particle components. Sentences have been rewritten.

In term of pollution from combustion process, it is necessary to focus on carbonaceous particles (OC and EC) since carbonaceous particles are the main constituents of the particle phase from combustion activity emissions.

Line 18-20. Sentence is awkward. "is" should be "are" See above comment on NMVOC. Particulate matter should lower case.

This paragraph was corrected

It is also interesting to study volatile organic compounds (VOC) because the emission factors of these components are not well-known despite their expected impact on air quality and climate through their

effects on ozone and secondary organic aerosols (SOA) formation (Matsui et al., 2009; Yokelson et al., 2009; Sharma et al., 2015).

Line 22. Acronym should be in parentheses. Work Packages should be capitalized.

This line was corrected

Line 26. Primary is redundant.

"Primary" was removed as suggested by reviewer.

Line 31-32.  Section 1 and 2 don't correspond to sections in the paper.  Please state whether this work is being presented in other papers and which measurements this paper will cover and which will be covered in other papers.

This paragraph has been rewritten. This work is only presented in this paper. For your information, two papers in the same area and linked to DACCIWA emissions are scheduled :
- One on a new regional inventory from 1990 to 2016
- Another on VOC ambient measurements compared to source measurements in terms of atmospheric concentrations

In section 2, this paper describes the materiel and the methodology used to calculate EFs of the main studied African emissions sources. Section 3 deals with the analysis of samples whereas section 4 presents the EF results of field measurements including a comparison with literature values. In this section, combustion chamber measurements of EF are also added.

**Page 4**

Line 7. Should it be Ward et al. 1993?

Yes, it is Ward et al. (1993)

Line 15. This equation is incorrect to me as it is currently written. I do not understand why the MCE was added to this equation; it adds unnecessary terms that make the equation no longer correct. If this is how the emission factors were calculated, I believe all the EF values in this manuscript need to be corrected by removing the MCE term.

If the authors feel this is correct, they need to provide an explanation and a reference where this equation was used for EF calculations.

Thanks a lot for this important correction. It was a mistake to include MCE values in this equation. Our values were all checked.

MCE will be used later to calculate emission values from EF and consumption values especially for wood combustion. Note that it is then interesting to also calculate MCE values from our field campaigns and this is the reason why we give details on MCE values.

**Page 5**

Line 3. List countries.

Done. The list of countries was added in the revised manuscript.

The first in March 2015 in Abidjan (Côte d'Ivoire), the second in July 2015 in both Abidjan and Cotonou (Benin), and the third in July 2016, also in Abidjan

Line 5-8. What is meant by "measurement points" and "samplings" - eight locations or eight tests from different plumes, or eight samples from the same plume?

It was eight locations with different sampled plumes: this part has been rewritten

In each eight different locations chosen to represent the combustion of waste diversity (dry, wet, old or fresh waste), trash burning plumes were sampled at "Akouédo" landfill, the largest (153 ha) and the official landfill site in the east of Abidjan District.

Line 10-11.  Please elaborate on "different characteristics".  Does "services" mean electricity?

Done. Hevea and Iroko are both hardwood but have different properties; for example, their densities differs (600 kg/m$^3$ and 650 kg/m$^3$, respectively).
Services means here bakeries, restaurant and other commercial services. This is clarified in the text now.

Line 12-14. More details are needed for these measurements. How much fuel was used? Was the fuel prepared, e.g. dried before use? Were any fuel properties tested such as moisture content? What were the cookstove/kiln characteristics, designs and models; show pictures/diagrams of these. Show sampling setups. Show location where these measurements were conducted on a map if they were field measurements. How exactly were the burn experiments initiated and conducted- what were the exact protocol? Please give this information for each emissions sources studied.

We do not estimate fuel quantities but we have used the carbon balance method. In this method we estimated the quantities of carbon released in the atmosphere during the combustion which quantities are related to carbon contained in the wood and wood amount burned. The fuel was not dried after use. The moisture content of Hevea wood (600 kg/m3) and Iroko wood (650 kg/m3) was not measured during field measurements but only during Edinburgh combustion chamber measurements. In addition, Hevea wood contain latex unlike Iroko wood. Locations where these field measurements were conducted were shown on a map in supporting information Figure S1. All the details are now given in the text or in the supplementary document.

P5-L30

How often was the instrument's calibration checked?  Was this done in the field ?

The instrument was calibrated in the laboratory prior to each field measurement, but not in the field. In addition, we used the difference between two values (before and during combustion) for EF calculation, so the zero calibration does not influence EF calculation. Qtrak reference are (Kam et al., 2011; Li et al., 2017), it was CO Benchmark monitor (Curto et al., 2018).

**Page 6**

Section 2.4. Misleading title. The combustion didn't occur in the chamber, the chambers were just used for dilution.  Why were these chamber experiments conducted? Were they part of this initiative? Please include rationale in the Introduction.

The title was rewritten to include the importance of combustion chamber measurements.

2.4 Measurements in the combustion chamber in the diluted wood combustion plume

Line 1. Why was air used? I believe heated air (oxygen) damages the sorbent.

Thanks for pointing out this mistake. Purified nitrogen for tubes conditioning was used and the sentence was appropriately corrected in the revised manuscript

Line 4. State the pump vendor/model.

Manual pump (Accuro 2000, from Dräger)

Line 5. Why was no size cutoff selected?

The instruments we have to conduct these field measurements do not allow us to have size cutoff

State quartz filter vendor PN, size

47-mm-diameter quartz-fiber filter (QM/A®, Whatman Inc.)

Line 9. Which sources?

Here, Hevea wood burning was the only source studied. This has been clarified in the revised manuscript

Line 14. Does this mean room temperature or lower? What is meant by "homogeneous"? There are gas and particles, thus it is not homogeneous. Do you mean well mixed? Which stove was used for the combustion – give more details or reference with details.

With "low temperature", means "a room free of solar radiation". A fan stirs the air in the chamber to mix it well. For that combustion we used a metal fireplace similar to the one used in the field as well as a system to convey the plume in the chamber. This part has been revised

In Lannemezan, the dark dilution chamber (10m x 4m x 4m) allowed measurement of concentrations at the absence of any solar radiation with no photochemistry. A fan stirs the air in the chamber to mix it well.

Line 24. What were the conditions in this study?

The conditions in this study were stated above in Page 6, Line 19-21

Hevea wood combustion experiments were also conducted using the FM-Global Fire Propagation Apparatus (FPA) at the Edinburgh University School of Engineering facility.

Line 28. This statement is misleading. It has not been made clear that these chamber measurements have already been discussed in detail in Haslett et al. Why are these measurements being discussed again in this paper? If the sampling details are given in this reference, they do not need to be repeated in detail here

In this study, we added other filter sampling to those made by the FPA. They are not described in Haslett et al., (2018). This sentence has been rewritten.

A large part of the chamber measurements have already been discussed in detail in Haslett et al., (2018). Such paper mainly focused in active analyser results but not on filter sampling. The configuration used is shown in Figure 3.

**Page 7**

Line 3. This is not a complete sentence.

This sentence was revised

The $CO_2$, CO and $O_2$ concentrations in the plume were measured at a frequency of 1 Hz at the exhaust tube by the FPA analysers using non-dispersive infrared techniques (Servopro 4200)

Line 10-14. This section describes the sampling media that should be discussed in Section 2.3.

This section was moved and rewritten in section 2.3.

Line 11. Please change "absorbent" to "sorbent".

This was done in the new version of manuscript

Line 12.  Please elaborate in more detail why these two types of tubes were used. Did these tubes capture different volatility ranges or different classes of VOCs? State vendor and part number for tubes or if hand packed, state so.

The methodological strategy selected was based on the measurement of the maximum range of VOCs for the largest number of sources. In this way, two types of sorbent materials and analytical techniques were applied during this study. Carbopack cartridges allow the sampling of a wide range of VOCs including hydrocarbons, ketones, alcohols and aldehydes, and all polar compounds within the volatility range specified. In the work of Detournay et al. (2011) it was shown that carbopack cartridges have the advantage of offering a specific trapping surface and the thermal stability that ensures a thermodesorption of 6-16 carbon atoms compounds. In the case of Tenax tubes, aromatics, non-polar compounds and less volatile polar compounds can be sampled (Ras et al 2009). The cartridges were provided by TERA Environnement and SUPELCO ready to use.

Moreover, this campaign was developed after other VOC campaign performed in Abidjan were ambient concentrations were analyzed. In that case, only Tenax cartridges were analyzed and fifteen VOCs were quantified.

Line 14. Where was LAMP located? Need to discuss how samples were transported and hold times between sampling and analysis for all samples. Has it been confirmed that stability times for all target compounds on the tubes exceed the hold times in this study? Were any quality control samples, such as field blanks or controls, included?

LaMP laboratory is located in Clermont-Ferrand, France. The samples were transported by plane after the field campaign. The holding time between sampling and analysising was less than 3 months. The stability time and the method reproducibility for the cartridges had been evaluated in previous studies (Detournay et al 2011). During these tests, the cartridges were injected with standard gas. Half of them were analyzed immediately after sampling and the other half 30 days later. The results showed excellent relative standard deviations (RSD) for aromatic compounds and alkanes (1%); and satisfying results were found for terpenes (5%) and oxygenated compounds (15%). More details are given in Detournay et al (2011). During field campaign several blanks were used in order to control the stability of the sorbent materials.

Line 21,27.  Please list the compounds that were measured and discuss why these compounds were selected. If they are listed in SI, note here. Were there overlapping VOCs? If so how did they compare between the two tube methods? Also provide method detection limits for each VOC and each instrument in SI.

Line 25.   Why were different types of tubes analyzed in different labs?   Spell out acronym.

The question was answered above (see P7-Line 12)

Line 26,27. State exact number of compounds. See above comment.

The question was answered in the revised manuscript and the required data was attached to SI material. The list of VOC quantified in each VOC group were detailed in Table S4.

The P. 7 L.21-27 was modified as such:

This method allowed the separation and identification of 58 compounds, from C5-C16 VOCs, including 7 carbonyls, 2 ketones, 12 terpenes and 6 intermediate VOCs (C11-C16). The application of both methods allowed the comparison of common compounds (benzene, toluene, ethylbenzene, m+p-xylene, o-xylene, trimethylbenzenes, n-heptane, iso-octane, n-octane, α-pinene, β-pinene, limonene, isoprene) and the analysis of analytical techniques performance.

The tubes were previously conditioned by flowing purified air through them at a rate of 100 mL min-1, for 5 hours at 320 °C using an adsorbent thermal regenerator. The quality assurance parameters of both methods (uncertainties and detection limits) are described in SI (Tables S1 and S2).

An also stated in P13 L2-4:

Volatile organic compounds were measured and emission factors estimated for the first time in West Africa. Fifteen common VOC species (C5 to C10) were identified and quantified from sorbent tube measurements and are reported in Table 7. The selection of these VOC was related to their identification by both analytical methods implemented during this study.

Line 28. This is misleading. For this study, only TPM is measured.

For field measurements only TPM was studied but for Lannemezan combustion chamber, PM10 and PM2.5 were also studied. This is clarified

Line 31, 32. This should be called elemental carbon instead of black carbon. Correct the manuscript accordingly. Also, these are particles and not aerosol measurements – please correct sentence.

This was corrected in the revised manuscript.

**Page 8**

Line 2. By "the thermo-optical method", this refers to the IMPROVE method? Isn't the method used in this study also a thermo-optical method? What was the rationale behind using this less well validated method as opposed to IMPROVE or NIOSH? Which samples were used in the comparison; were they from this study or other samples? Was the analysis done on the same instrument? What instrument was used, and how was it calibrated?

These questions are very important questions, we agree. The well validated thermal method based on Cachier et al. (1989) was used for all samples due to the high load of the filters, difficult to analyse with thermo-optical methods. However due to different method uncertainties, a comparison between the used thermal method and the IMPROVE thermo-optical method, was conducted for 10 slightly loaded samples.

Thermal method measurements are done with G4 ICARUS HF carbon analyser. The analyser calibration is checked before each series of analyses by analysis of filters each containing a sucrose solution with a known concentration.

Line 4. What is considered suitable?

The purpose of this comparison was to show that the thermal method used in this study gives roughly the same results than the IMPROVE method. Considering filter high loads, the thermal method was considered appropriate to be used.

Line 15-18. "Residential sources" is too vague. Need complete sentences between semicolons.

Sentences were completed

Field measurements allowed mean values EFs determination for residential fuels fires (charcoal burning (CH) and fuel wood burning (FW)) to be obtained. Mean EFs values for charcoal making fires (CHM), for open trash burning fires (WB) and for vehicle exhaust (car, bus, truck, light duty vehicles, two-wheeled two-stroke and four-stroke vehicles) by energy source (Diesel and Gasoline) and by age group (recent (less than 10 years old) and old(10 years and over)) were also obtained.

Line 20. Please change "aerosol" to "particulate" or "particle". This was done

Tables should be numbered in order of when they are first mentioned in the text. This Table should be Table 1.

This sentence "Speciated VOCs list is shown in Table 7" was moved to VOCs results section.

Only 15 VOCs are listed.  Why is only a subset of compounds listed here?

Here we have listed the species analyzed by both laboratories (Lamp and IMT Lille Douai).

It is mentioned that three tests per source were performed – how many samples per test were taken? Generally, one filter and one sorbent tube were sampled per test.

Please list exactly how many tests (and samples taken per test) were conducted for each source. This could go in the SI.

Table S3 of the Supporting Information list the number of samples taken per test for filter and sorbent tube.

Line 21-22. Why geometric mean? Which sectors are being referred to?

It is a mistake, we are sorry, weighted mean was used for mean EFs. This was corrected.

Line 26. Fossil should not be capitalized.

Done. This was is corrected in the new revised manuscript

Line 28. Please state the biofuels.

Done.This part was reformulated and moved in section 4.1.1

**Page 9**

Section 4.1.1. I suggest removing Section 4.1.1 as much of it is quite predictable and not scientifically interesting, and instead discuss MCEs along with the EFs in other sections.

Follow Referee#1 remarks section 4.1.1 was removed and MCEs has been discussed with the EFs in each corresponding results sections

Line 1.  Why are MCEs summarized in a Table and Figure – this is redundant.  The MCEs can easily be summarized in Tables 2 and 3 with the emission factors.  Modified combustion efficiency should be lower case.  Both MCE and $CO/CO_2$ are both combustion indicators – it is redundant to discuss both.

Table 1 was deleted and MCEs has been summarized in Tables 2 and 3 as recommended by the referee #1.

Line 2-4. The meaning of this sentence is not clear. Please clarify.

More details have been provided in the revised manuscript.

Line 14. BC/TC ratios are mentioned here before showing any results on their emission factors.  This relationship with MCE should be brought up during discussion of "BC" emission factors.

As mentioned above, section 4.1.1 was rewritten in each of the corresponding results sections as recommended by the reviewer.

Line 19.  Title is too vague.  Residential heating?  Residential cooking?  Is charcoal making a residential source? Title was replaced by "Particulate EFs of biofuel combustion sources".

Charcoal making is not a residential source but we put it in this 'activity' because the product resulting from charcoal making is used in this subsector.

For these subsections (4.1.2-4.1.4) only particulate EFs are discussed, so make that clear in the subsection titles.

This section was reorganized following the reviewer's comments. Thank you

Line 20, 22. EF should be plural – EFs. Please check the entire manuscript for this error. Please reference Equation 1 consistently.

That was corrected always in the revised manuscript. All equations have been referenced correctly.

Line 21. Table should be capitalized and it should be Table 2. Please check the rest of the manuscript for this error.

All tables in the revised manuscript were referenced and written correctly.

Line 22. "Dry matter" is misleading. The calculation is per kg fuel burned not dry matter. Please make this clear.

Thank you, we agree with this remark, it was corrected throughout this revised manuscript where dry matter has been replaced by fuel burned.

Line 23-25. Is there any evidence behind these assumptions?

This was really an assumption, since it is impossible to exactly quantify the relative use of the different woods

it has been replaced by :

To have a mean EFs for wood cooking fire, we averaged the EFs of the two types of wood studied here (Iroko and Hevea)

Line 26-28. This is a contradictory statement. Are they in agreement or higher than literature? What fuels did these other studies test? Have emissions from Hevea, Iroko or any other African-specific fuel been measured previously?

Follow the comments of referee #1, references of EFs measurements not performed in Africa has been removed. Previous measurements of wood combustion emissions in Africa did not explicitly mention the wood species studied but specified that they were local species.

**Page 10**.

Line 1. This is too vague. What studies?

These are studies by Brocard (1996) and Roden and Bond (2006). This line has been reformulated

It may be seen here, our EF(EC) for charcoal cooking fires are 3 times higher than those reported by Brocard et al., (1996) and Roden and Bond, (2006) but the same order of magnitude to those used by Liousse et al., (2014). On the other hand, the EFs are almost the same for OC.

Line 2-4. I do not agree with this argument. If carbon is left in the fuel, it is not burned and not emitted - so it is not included in "mass of fuel burned" using the mass balance method.

Since an important of burned carbon is not volatilised but mainly left in charcoal and in pyroligneous liquid, we cannot use the mass balance method for CHM EFs as done for wood and charcoal burning.

The carbon balance method cannot be used directly to calculate charcoal making (CHM) EF (Bertschi et al., 2003) in the same way that we did to calculate wood and charcoal burning EFs. Indeed, during CHM process, part of the burned carbon is found in charcoal, ash and in the pyroligneous liquid. Then, less than 50% is emitted into the atmosphere.

Therefore, I do not believe adding this additional multiplication factor is correct.

As MCE was removed to Eq.1, EF(g/kg carbon) was multiplied by the fraction of carbon emitted as $(CO_2+CO)$ to obtain EF in g/kg wood burned (Cachier et al., 1996).This was corrected in the revised manuscript

Thus, in order to obtain an EF in g/kg of wood burned, 0.35 multiplied the EF in g per kg of carbon (Cachier et al., 1996).

Line 11. This should be Table 2. Please check entire manuscript for this error in Table numbering.

This was corrected in the revised manuscript

Line 13. This statement is contradictory, as it was earlier stated that CHM conditions in Cachier et al. were similar to this study.

Thank you. The statement has been removed.

Line 18. Recent and old are not age groups and are thus meaningless adjectives. Please state model year groups.

Vehicles were classified into two categories: old (10 years and over, meaning vehicle model year after 2006) and recent (less than 10 years old, vehicle model year prior 2006) vehicles. Another classification criterion in addition to vehicle age was visual observation of the opacity or non-opacity of the exhaust gas.

Line 19-22. Which "literature values" are being discussed in Line 19? Why were these particular studies selected for comparison given the large number of vehicle emission studies available? Do they have any applicability to West African vehicle fleet? If not, then what is the reason for the literature comparison? State units for EF values in Line 19.

Based on the literature review and the knowlegde of Africa, up to now, there are no studies on emission factors of pollutants emitted by road traffic specifically measured in Africa. Also, measurements of emission factors in other parts of the world are mainly used to compile emission inventories in Africa. However, we agree with the referee's comment and as a result, we have removed the comparison with those EFs that are not measured in Africa. Thus, we focused the comparison on the EFs values used in the global inventories for Africa (Bond et al., 2004) and the regional inventory for Africa (Liousse et al., 2014).

Line 23. Please explain the meaning of "coherent with Fig 4".

Thank you, this has been reformulated.

Figure 4 does not show this statement to be true for diesel EFs in this study, if I understood what was mean.

Figure 4 shows that BC/TC ratio is very high for the measurements performed on diesel sources showing that OC is much lower than BC in that case. However, this figure has been removed following the reorganization of the manuscript.

Line 27-31. This is probably one difference of lesser importance than vehicle age and lack of emission controls. Therefore, it is unclear why it is emphasized over other more important differences. Provide reference for evidence supporting these statements/assumptions in Lines 28-31.

We agree that the measurement method probably induces less important differences than the age of the vehicles and the lack of emission controls. Therefore, we have focused more on the vehicle age and the lack of emission controls.

What is meant by "park vehicle"?

It means fleet of vehicles. This has been corrected

Lines 32-33. Include units.

Thank you, unit was added

**Page 11**

Line 3. Why compare to HDDVs? They are not relevant.

This comparison was removed

Line 4. Please report all EFs for individual vehicles in the SI. I don't see how these average values are helpful and why they require a separate table. How are these values expected to be used?

All EFs for individual vehicles were reported in the SI. For emissions calculation, national value of road gasoline/diesel consumption are mostly used due to the lack of detailed information per vehicle class in African countries. Thus, average EFs values allow to have data specific for an average fleet (either gasoline or diesel) very helpful for road emissions calculation in these countries.

Line 8. Why are these values compared to European inventory? What is the usefulness if this comparison?

We have compared these values to COPERT EURO values since inventory developers building African emissions inventories mainly use these EFs values. This comparison gives an idea of the uncertainty that is committing by using these EFs for inventories in Africa. However, we have removed this comparison because it is not relevant as mentioned by referee # 1

Line 11. This statement is misleading. Approximate agreement of OC/EC ratios does not justify this general statement.

This was removed

Line 13-15. This is the most relevant literature comparison I've seen in this section, so this needs to be discussed in much detail.

As mentioned above, a comparison with EFs used by Liousse et al., (2014) and Bond et al., (2004) was discussed in detail in the new version of the manuscript.

For gasoline vehicles, EF(EC) value of our mean road equivalent vehicle is higher than the upper limit value used by Bond et al, (2004), which is the value used by these authors for countries where there are the most super emitters as in West Africa. Unlike EF(EC), our EF(OC) is within the range of values given by Bond et al. (2004) however it is lower than the upper limit given by Bond et al., 2004. For diesel vehicle, while our EF(EC) value for mean road equivalent vehicle is close to the upper limit given by Bond et al., (2004), our EF(OC) is higher than their upper limit value. In addition, the EFs values obtained in this study for gasoline are higher than those observed in Liousse et al., (2014) (4 times higher for EC and 2 times higher for OC), while they are slightly lower for diesel (Table 2). This shows that the use of Liousse et al., (2014) EFs underestimate EC and OC emissions for on-road gasoline motor comparing to what would happen if using our EFs values whereas of the same order of magnitude for on-road diesel motor.

Line 16. Weren't all the vehicles classified the same way? Vehicles were all classified in the same way

EFs of two-wheeled (TW) vehicles were also classified according to age (old and recent), and engine type (two or four strokes).

Line 27-29. See comment for line 13-15.

Road Mean TW EFs comparison with Assamoi and Liousse, (2010) and Bond et al., 2004 EFs' values, for EC and OC, have also presented.

Mean road equivalent two-wheeled vehicle EFs'obtained for EC and OC (Table 3) are both very close to the values used by Assamoi and Liousse, (2010) which is the only regional two-wheeled vehicle inventory specific to Africa. These values were used in Liousse et al., (2014). Also, EF(EC) and EF(OC) mean values from this study are both above the upper limit of the TW EFs given by Bond et al., (2004) which correspond to the TW EFs that these authors consider for Africa. Comparison of these new values with those used in the Bond et al., (2004) inventory shows that this global inventory underestimates EC and OC emissions from two-wheeled vehicles in Africa, particularly in West Africa.

**Page 12**

Line 2. Instead of "bad" I suggest "less efficient".

Done. Thank you

Line 11-13. Were these different phases observed during the same test/sample or different tests?

These different phases were observed during each test/sampling with some predominance of one of these phases from one test to another.

Can you include a Figure showing real time $CO_2$, CO measurements during a burn to demonstrate this? Please expand in greater detail.

[Figure]

**Figure S4: Trash combustion characteristics**

During the burning, the heat increased the air temperature and decreased the air humidity. Thus, the air temperature was higher in flaming-dominated combustion conditions than in smoldering-dominated combustion due to higher heat released during the flaming phase.

Line 23. Why are only these VOCs shown? Did they have the highest EFs for all sources? Where are the emission factors for the other VOCs?

As stated before, the selection of these VOC were related to their identification by the both analytical methods implemented during this study. This selection allowed the comparison of different analytical method from duplicates obtained during field campaign.

Line 25-27. Aren't all VOCs important for atmospheric reactivity to some degree?

More clarification are given in the revised manuscript following reviewer comments.

Line 29. Given that no background samples are taken, how is it known that these are not solely from biogenic emissions? Please discuss.

The measurements for the emission factor estimation were performed directly at the source (at a distance of 1-1.5 meter) in order to guarantee only the contribution from the specific source. In parallel, ambient measurements were performed in different sites (not discussed here), allowing us to compare later the VOC profiles between ambient and emission sources.

Line 31-32. These analytical uncertainties between methods needs to be discussed in greater detail in the Methods section. Please clarify the statement "different sources associated to the emission sector analysed".

Global uncertainties associated to each compound and to the both analytical method used were calculated and are detailed in SI

**Page 13**

Line 1. What is meant by "most important"?

Thank you. The text was changed to be clearer

Line 10. What do these values represent?

The text was revised and statement made clearer

Line 20.   Since a full VOC target list has not been given, it is impossible to evaluate whether enough compounds were measured to be representative of the different species groups.

The VOCs species considered in each VOC group were detailed in table S4. Considering that this study analyzed for the first time VOCs emissions in West Africa, we integrated the data in VOCs groups as suggested in the bibliography. This method allows us to evaluate discrepancies and commonalities in order to access more accurately to VOC emission profiles and their associated uncertainties.

Line 26. If the aldehydes were a large contribution, why were they omitted from Table 7 or discussed earlier in this section?

As stated in the section 4.1.5 the selection of the VOCs reported in Table 7 are related to their identification by the two analytical methods used. This was our criteria in order to provide a standard deviation to the corresponding EF.

Line 29.  The relevance of these particular studies for comparison needs to be explained. They do not seem to be at all applicable to this work, so this comparison is not helpful.

Since VOC species from direct emission sources were measured for the first time in Africa during our study, there are no other African reference to compare the results obtained. In the same way, and to analyze the contribution from African sources under study here, we compared them with the same

sources in other worldwide places. Thus, this exercise allowed us to understand the magnitude of African emissions on the regional atmosphere and to quantify the impacts of these emissions.

**Page 14**

Line 1. I am not convinced that this is true based on the discussion above.

The text was revised and statement made clearer

The presence of isoprene and monoterpenes in WB emissions is also observed in the literature. However, in this study, their levels are as significant as the ones of aromatics compounds.

Lines 5-10. This should be moved to the Methods section on EF calculations.

This part (line 5-10) was moved to method section

Line 14.  There are many possible reasons for these discrepancies. Were the fuels burned using the same cookstoves under the same protocol? Were the MCEs similar? Were the fuels of the same moisture and carbon content? Perhaps the sampling methods/particle losses were different? Dilution can certainly affect OC (and this should be discussed further), but how will it affect EC EFs? Please elaborate.

Yes, there are many possible reasons for discrepancies between field and combustion chamber (CC) EFs values but the main factor was plume dilution. MCE of both measurements type are different, 0.76-0.92 and 0.97-0.98 for field and CC, respectively. This confirms that the combustion conditions are different from field at CC. Although, we have used the same species of wood (hevea) for measurements type thus, it has the same carbon content.

This sentence was reformulated as:

This important difference between field and combustion chamber results may be linked to many factors such as the high dilution of plumes occurring in combustion chambers and on other hand to different combustion conditions (different MCE)

Section 4.2.1.   These measurements have already been compared in the previous paragraph. Why is there subsection here in the middle of this continuing discussion on the topic?

This section was restructured to avoid repetition as suggest by reviewer. Thank you

Line 26. If CO/CO2 ratios are that different that is a measure of combustion efficiency, then aren't the combustion conditions also very different between field and lab? Why are BC/TC ratios used as a combustion efficiency indicator instead of MCE values?

As the MCE, the CO/CO2 ratio is a combustion indicator and therefore has different values in the field and in combustion chamber. BC/TC was not used as combustion efficiency indicator but it was used here to show the predominance of main combustion phases (flaming and smoldering). Therefore, BC/TC was removed and this part was edited.

Line 31. Filter method in this study measured EC not BC. There could be differences due to the different methods that should be discussed here.

This part was discussed and added in the new version of manuscript

**Page 15**.

Section 4.2.2.  What can be said about size distributions from the other emission sources in this study? Please discuss this issue for each emissions source and give appropriate references.

In this study, only wood combustion EF measurements per size class were done in combustion chamber.

Line 7. Diesel engines are not relevant.

This line was removed as suggested by reviewer

Line 17. This sentence is too vague.

This sentence was reformulated

During field measurements, several tests were performed per sources studied here in order of getting the more representative EFs for each source.

Line 20. Please state that it is particle mass.

This was corrected and the sentence reformulated

Particle EFs for biofuel burning are comparable to the range of those found in the literature (measurements and values used in inventories in Africa) excepted EF(OC) for wood burning that emit more particles than charcoal burning.

Line 21. What EFs are being referred to? None of the EFs were shown as a function of vehicle age or maintenance, so this conclusion has not been confirmed by the previous discussion.

In section 4.1.3, we have discussed of EFs for two vehicle age classes (recent and old) but not for its maintenance, the sentence was reformulated and made clearer.

Line 24. "Published EF values" for what specific source?

These are the traffic sources for diesel and motor gasoline. This line was reformulated.

In contrast, particles EFs for recent (under 10 years old) vehicle models are slightly the same to published EFs values for gasoline and the same order of magnitude for diesel

Line 26. EFs of all VOCs have not been provided, so there is no evidence given that this statement is correct.

The sentence was reformulated and made clearer.

Moreover, the EFs of more than 50 VOC have been determined for the first time in West Africa and integrated to the GEIA VOC groups being available for the development of emission inventories.

Line 30-32. The meaning of this sentence is unclear.

The sentence was reformulated and clarified.

**Page 16**.

Line 1. State the fuel used for this measurement.

In combustion chamber Hevea also know as "rubber tree" was the alone studied species. This was reformulated.

In the combustion chamber measurements, the EFs of the particles for Hevea wood fire per size class show that EC is mainly in the fine fraction

Table 9. Please state the emission source/fuel burned in the caption.

The emission source in this caption was wood burning and the caption was completed.

Table 1: Relative contribution of EFs (BC) and EFs (OC) for wood burning per size classes to total size.

**Referee #2**

**Anonymous Referee #2**

This work presents the Emission Factors (EFs) from fuel combustion in West Africa estimated from field and combustion chamber measurements. EFs were estimated for black carbon (BC), primary organic carbon (OC), total particulate matter (TPM), and 50 non-methane volatile organic carbon (NMVOC) species. In addition, measurements in combustion chambers were used to estimate particulate EFs by size, namely for PM10, PM2.5, PM1 and PM0.25. This work was conducted within the framework of the DACCIWA (Dynamics-Aerosol-Chemistry-Cloud Interaction in West Africa) FP7 program. Field measurements were conducted in two places of West Africa, namely Abidjan in Ivory Coast and Cotonou in Benin, while measurements in combustion chambers were conducted in Toulouse, France, and in Edinburgh, United Kingdom. The considered emission sources are wood and charcoal burning, charcoal making, open waste burn ing and vehicles including cars, trucks, buses and two-wheeled vehicles. As mentioned by the authors in the introduction, one of the sources of uncertainties in emission inventories are uncertainties in the emission factors and therefore research contributing to reduce these uncertainties are necessary to better quantify the impact of given source sectors on air quality and climate. However, although I recommend this paper to be published in ACP some important aspects need to be addressed before its publication.

General comments

1. The authors mention the role of uncertainties in the uncertainties of emission inventories (EI) in the introduction, however they do not address this issue again in the rest of the manuscript. How does this work contribute to reduce uncertainties in current EI in Africa? What are the main sources of uncertainties in EF in the region?

Thank you for reporting this deficiency, it has been corrected in the manuscript. Emission inventories (EI) uncertainties are due to both EFs and activity data uncertainties, thus this work will contribute to reduce uncertainties in current EI in Africa by taking into account African specific EFs. The main uncertainties of EF in West Africa, revealed by this study, belong to the traffic sector composed of old vehicles and using poor quality fuels, but also in the domestic sector with the use of wood and charcoal.

2. After reading the article it is not clear to me what the real contribution of this article is. The authors make an attempt of this in the last paragraph of the conclusions but not supported in the text. The authors should make an effort and discuss the implications of their results for current Emission Inventories. Are emissions currently over or underestimated if these new EFs are considered?

The implications of using the results of this study for current emission inventories were discussed through the new version of the manuscript as suggested by the referee. For example, the use of these new EFs has shown that current emissions are underestimated for OC, mainly due to wood fire OC EF's (2.7 g/kg to 11 g/kg) which is the main source of carbonaceous particulate emissions in West Africa.

3. It is not clear to me that the title is the most appropriate considering the scope of the study. Measurements are conducted only in two places of tropical West Africa but the authors claim (not explicitly) that the results are valid for Africa. How representative of other countries in Africa and/or West Africa are these results? Can they be extrapolated for the entire region? The authors show with their results that EF are sensitive to multiple factors therefore applying these emission factors to other countries is not straightforward. The scope of the study should be consistent with the title and the content of what is presented. It is not only a matter of the title but also how the data are presented and their representativity.

First, the title has been modified "Particle and VOC emission factor measurements for anthropogenic sources in West Africa". We think that these results are representative of other countries in West Africa because this region have slightly the same tree species that is used for energy purposes such as wood burning and charcoal making. In addition, all countries in this region and most of the sub-saharan Africa, are characterized by older vehicle fleet imported from western countries to be used for traffic and/or personal cars. These vehicles often use slightly the same quality of diesel and gasoline.

4. The authors are not thorough when presenting the results and formulation is unclear in certain places. Authors should review the manuscript for consistency and improve formulation. For instance, in page 9 the numbers provided from lines 5 to 7 do not match the numbers given in the table indicated in the text. Also in page 11, line 18, the values in the text are approximated to one decimal while in the table it is with two. Authors should review the text and correct inconsistencies between the data provided in the text and the ones given in the tables and figures.

All these inconsistencies were corrected in the revised manuscript.

5. The authors make an attempt to put the results in perspective in the last paragraph of the conclusions but without elaborating on it in the text. The authors should elaborate on the impact of the EFs and provide more evidence on the importance of their findings for current and future emission inventories.

The implication and impacts of the use of new EFs in emission inventories are now discussed in the revised manuscript

6. Authors should make sure that all methodological description is presented and described in section 2. At present, section 4 also includes important methodological aspects in terms of composition of sources and how the data are combined to obtain total EFs per source. This should be moved to section 2 and more information should be presented. It should be made clear to the reader how the data are aggregated.

All methodological aspects of section 4 were moved in section 2. Thank you

Specific comments

**Page 3**, lines 18-21: formulation is unclear and should be improved.

Thanks. We have changed the sentence:

It is also interesting to study volatile organic compounds (VOC) since the emission factors of these components are not well-known despite their expected impact on air quality and climate through their effects on ozone and secondary organic aerosols (SOA) formation (Matsui et al., 2009; Yokelson et al., 2009; Sharma et al., 2015).

**Page 3**, lines 22-25: Unnecessary information is provided on works packages of the DACCIWA project but no general information of the project itself is provided. I suggest removing the unnecessary information and include a short description of the project and then linking it to the work presented, without necessarily tying it to a WP.

This was done:

Our study is included in the frame of the Dynamics-Aerosol-Chemistry-Cloud Interactions in West Africa (DACCIWA) programme (Knippertz et al., 2015). DACCIWA aims to quantify the influence of anthropogenic and natural emissions on air quality, clouds and rainfall over southern West Africa and assess their impact on human, ecosystem health, and agricultural productivity. One of its aims is to develop an emission inventory of anthropogenic sources specific to this region. In this framework, several campaigns to measure EF were performed.

**Page 3**, lines 31-33: Section numbers are not correct, in addition the paper has five sections and only two sections are presented and described. The authors should complete this paragraph and make sure that section numbering is consistent.

This was corrected:

In section 2, this paper describes the materiel and the methodology used to calculate EFs of the main studied African emissions sources. Section 3 deals with the analysis of samples whereas section 4 presents the EF results of field measurements including a comparison with literature values. In this section, combustion chamber measurements of EF are also added.

**Page 4**, line 1: remove "in the frame of the DACCIWA WP2" since this information has already been provided and information on the WP is not really informative and relevant.

This was corrected:

"in the frame of the DACCIWA WP2 " has been removed and the sentence reformulated as:

Two types of measurements were carried out in this study for emission factor measurement experiments: field measurements for all studied sources and combustion chamber measurements for fuelwood.

**Page 5**, lines 4-7: formulation is unclear and should be improved.

This was reformulated:

(1) Open trash burning: In each eight different locations chosen to represent the combustion of waste diversity (dry, wet, old or fresh waste), trash burning plumes were sampled at "Akouédo" landfill, the largest (153 ha) and the official landfill site in the East of Abidjan District

**Page 5**, line 31: What is the unit of the provided uncertainty, is it % or ppm?

The unit of the provided uncertainty was %. This was corrected in the revised manuscript.

**Page 5**, lines 2-23: The description of the field measurements needs to be improved. It is unclear weather the same measurements were conducted in each on of the 3 field campaigns or if they were different. For instance, when it is said that eight measurements were carried out at Akouedo was it in each one of the three campaigns or only in some of them. Please clarify, manuscript should be clear in terms of the measurements that were conducted in each campaign.

The description of the field measurements was rewritten by adding the period and number of sampling per site for each source studied. The description of the field measurements have been rewritten and clarified.

**Page 5**, line 27: Provide a reference of the instrument QTRAK-7575.

Qtrak reference is (Kam et al., 2011; Li et al., 2017), it was CO Benchmark monitor (Curto et al., 2018).

**Page 6**, section 2.4: Section 2.2 indicates that two types of African hardwood were tested in combustion chambers, namely Hevea and Iroko, however in section 2.4 only Hevea is mentioned and it is not clear whether Iroko was tested at all. Was Iroko included in the study? Results corresponding to wood burning correspond only to Hevea? Please clarify.

Section 2.2 deal with field measurements only where the two woods were studied, whereas in combustion chamber only Hevea wood also called "rubber tree" was studied. Thus, the EF results in the combustion chamber correspond only to the combustion of Hevea, whereas in the field, Hevea and Iroko were both studied. This has been clarified in the section 2.4.

**Page 7**, line 9: Section 3 should in fact be part of section 2. Why do the authors consider it should go in a section by itself?

We separate the measurements methodology from the analyses. This was done in the purpose to provide more details and avoid very long sections

**Page 7**, line 26: . . .in Detourny et al. (2011) and Ait-Helal et al. (2014).

This was corrected. Thank you

**Page 8**, line 20: First table mentioned in the text is Table 7! But no mention was done before to the previous 6 tables. Numbering of the tables should be done in order they are referenced in the text.

All tables are now correctly numbered in the manuscript

**Page 8**, lines 20-21: What exactly is meant by this? Does it mean that in addition to the number of measurements indicated in section three additional measurements were conducted to reflect reproductibility or are these three-measurement parts of the total number of measurements conducted? This should be made clear and should be included in section 2 and not in the results and discussion section.

It is not a three additional measurements. We have made this clarification to emphasize the minimum number of measurements per source. This part was clarified in section 2.

**Page 8**, lines 21-22: What are the arithmetic and geometric method the authors mentioned, is it just averaging? Again, this should be made clear and included in section 2 where the measurements are described. See general comment made above.

This was corrected, mean EF was just a weighted average. This part was also moved in section 2 in the revised manuscript.

**Page 8**, line 26: DL and MO have not been defined in the text. They are defined in the caption of figure 3 but should also be defined in the text.

This was corrected

**Page 8**, lines 28-29: How was the MCE of biofuels from 0.6 to 0.9 from Iroko wood obtained? Was it from the combustion chambers or from the literature? This should be clarified.

This result has been obtained from field measurements. This was reformulated and clarified in the revised manuscript

**Page 9**, lines 1-2: the $\Delta CO/\Delta CO2$ ratio has not been introduced as a quality indicator before, the statement should be reformulated.

This was corrected in the revised manuscript

**Page 9**, lines 2-4: remove the coma after 0.32 and replace "showing" with "show".

This was corrected

**Page 9**, lines 5-7: Values provided in the text do not correspond to values in Table 1!!!

This needs to be corrected and the authors should make sure that the numbers given in the text are consistent with those in the tables and figures.

This was corrected throughout the revised manuscript

**Page 9**, line 15: Three sources are mentioned but only two symbols are provided in parenthesis. Furthermore, the authors make a too simple analysis of the results presented in Figure 4. Above 0.2

BC/TC one could agree with the authors that the larger the ΔCO/ΔCO2 ratios the smaller the BC/TC ratio. However, below 0.2 BC/TC the data suggest that regardless of the ΔCO/ΔCO2 ratio, the BC/TC ratio is mostly constant. The authors should elaborate on these two regimes and explain the reasons behind it.

We agree with the referees' comments, but following the reorganization of the manuscript suggested by the referees, this part has been deleted including Figure 4.

**Page 9**, line 24: What calculations do the authors refer? How were these calculations made and by whom? Authors should provide more information about this.

We agree with referee comments, Here we make an assumption about wood species used for cooking and other residential energy uses. This was corrected

**Page 9**, line 27: Hasn't there been anything more recent than the references provided in this line? The authors should look for more recent EFs estimates.

A recent reference Akagi et al., (2011) was added and this part was reworded

**Page 9**, lines 27-28: The literature review of EF in West Africa is not thorough enough to make this kind of statement. Make the statement relative to the studies included in table 2 and not general.

It refers to the EFs literature discussed in this study. This was clarified,

**Page 9**, lines 31-33 and Page 10, line 1: Very little is said about Charcoal EF in contrast to wood and charcoal making. The authors should elaborate more on Charcoal. Also, the authors should make it a separate paragraph from wood and not split it on two paragraphs as it is now.

EF's results for charcoal cooking fire were more discussed and now given in a paragraph separated from the wood.

**Page 10**, lines 4-5: Put parenthesis after years of publication for each reference.

This was corrected

**Page 10**, line 19: Again the values given in the text do not correspond to values provided in Table 3 and here again it's a matter of significant numbers considered in the text and in the table. The Authors should check the paper for consistency and use the same criteria with regards to significant numbers when presenting results.

Sorry again. This was corrected along the revised manuscript

**Page 10**, line 19: I do not agree that the estimated EF(BC) for new LDGV is within the range of literature values. It corresponds in fact to the lower limit of the range of values provided in the table. The authors should make the statement consistent with the data in the table. Also, it is unclear whether the literature values presented in the manuscript are from the same country or region or from elsewhere in the world. Although the measurement method is presented, not the country where the study is conducted and this information should be provided or at least considered when comparing the estimates.

We agree with referee comments. The literature values presented in the manuscript come from elsewhere in the world. The discussion on this part was rewritten with more emphasis on comparison with the values used in the inventories for Africa

**Page 10**, line 30-31: Where are these percentages of old and recent vehicles taken from? Are they based on statistics from the literature, governmental documents, etc? The authors should explain where these numbers come from. Furthermore, this should be included in section 2.

"To calculate the mean EFs for light-duty and heavy-duty vehicles (Table 3), we assume that the park vehicle is constituted by 60% of old vehicles and 40% of recent vehicles".

These percentages of old and recent vehicles were taken from official documents of Ministry of Transport in Côte d'Ivoire.

**Page 11**, line 7: Again where are these numbers of 77% of light duty vehicles and 23% heavy duty vehicles? Same as comment before, these numbers need to be justified somehow and also moved to section 2.

These percentages also were taken from official documentsof Ministry of Transport in Côte d'Ivoire.

**Page 11**, lines 8-15: Again values presented in the text do not match those in the corresponding table referenced in the text (Table 4). Furthermore, the data are compared to different COPERT EURO Standards without even presenting them and explaining them. Furthermore, why are these data included in table 4? I would strongly suggest the authors to rewrite this analysis including the different COPERT values in table 4.

The COPERT EFs are given on the one hand for particles with an associated uncertain EC, OC conversion factor and in the other hand in g/km travelled not directly in g/kg of fuel. Moreover, these measurements are not typical of vehicles in Africa. For these reasons, we have decided to remove this comparison from the document and focused comparison on EFs used in inventories for African.

**Page 11**, line 18: Again value of 26.0 +- 1.10 g/kg given in the text is not the same as the one given in table 5 (25.71 +- 1.1). If the authors decide to use a certain criteria of significant numbers for the table, they should use the same for the data presented in the text. There is absolutely no reason why numbers should not be exactly the same. Again, please be consistent!

Thanks for pointing this out, the text, table values and the significant number of digits have been harmonised throughout in the revised manuscript

 **Page 11**, lines 19-20: What exactly is meant in the sentence "The same difference is observed for old. . .". The difference in EFs (OC) in fact for old cars between two and four stroke is much smaller than for recent ones. The authors should correct the analysis and make it consistent with the data presented by them.

This part was rewritten:

For recent TW two-stroke engines, EC and OC EFs are 2.26 ± 1.40 g/kg and 25.71 ± 1.10 g/kg respectively, while, 0.11 ± 0.01g/kg and 0.45 ± 0.13 g/kg are found for recent TW four-stroke engines. In addition, for old TW two-stroke engines, EC and OC EFs are 3.45 g/kg and 124.21 g/kg respectively, while, 3.66 g/kg and 25.46 g/kg are found for old TW four-stroke engines (Table 3). This implies that, TW two-stroke engines globally emitted much more carbonaceous particles (especially OC particles) than TW four-stroke engines.

**Page 11**, lines 20-21: The statement that two stroke engines emit more OC than four stroke engines is made only for old vehicles, why is not the same analysis done for new vehicles?

This statement was made for both old and new TW vehicles and this part has been rewritten

Also, the data to support that claim is not the OC/BC ratio but the actual EF, authors should correct this. This was corrected

**Page 11**, lines 27-29: Where are the percentages of two and four stroke engines taken from? As before, these numbers need to be justified and also be moved to section 2. Also, the Table 5 is referenced here, but shouldn't it be table 4? Finally, rather than saying that the values of this work are in agreement I would suggest to reformulate and say they are comparable.

Here we made an assumption, since we could not find data on the percentages of two and four-stroke engines. This was reformulated and moved to section 2. This part was reformulated following referee suggestions.

**Page 15**, lines 21-22: Authors conclude on the dependence of EF from traffic to vehicle age and maintenance, although the latter was not included in any of the analysis presented. The authors should base the conclusions to the results presented in the article. I suggest maintenance is removed from the conclusions or the corresponding data to support that claim are provided.

We agree with this comment, the maintenance was removed from the conclusion and this part has been rewritten.

EFs for fossil fuel burning in traffic are strongly dependent on vehicle age. The older the vehicles, the higher EF values for carbonaceous particles are and are sometimes up to 100 times higher than the EF values found in the literature generally for recent vehicle.

Page 16, line 6: Authors claim that EFs obtained are representative of African sources but do not provide any evidence or analysis of this in the paper nor a reference to support this claim. I suggest the authors provide some evidence to support this statement or remove it from the conclusions.

This sentence was removed from conclusion

**Page 16**, lines 7-9: "This unique database. . ." why is the database unique? No mention to its uniqueness has been made before.

We said that this is unique because many of the sources studied here have not been studied in Africa before, which led inventory developers to use EFs from elsewhere measurements (Europa, America, Asia, Laboratories, etc.).

How will it improve emission inventories in Africa? What will be the impact of these database on emission estimates? Again, no mention of the impact of these EFs on emission inventories is made in the text to support this.

Much of the uncertainty in emission inventories comes from EFs and activity data. The use of EFs specific to Africa, sometimes very different from those measured elsewhere, would reduce emissions uncertainties.

Finally, how will this new EFs help decision makers? Wouldn't this usually be come from new emission inventories? These kind of statements, although tempting should be supported by the facts presented in the text which I believe are not. Please reformulate.

This was reformulated

This original database will be useful for improving and updating African emission inventories allowing better assessments of climatic, air quality and health impacts. The emissions obtained with these new EFs in this study can also help to identify key source categories. This could help policy makers implement new policies to mitigate emissions from major emitting sources.